# A Guide to Training Consistency Models

## Abstract

While the theoretical foundations of consistency models are well-understood, their practical implementation is often hindered by complex and entangled training pipelines. The interplay between critical components is not always transparent, making systematic improvement difficult. To address this, we propose a practical training playbook for consistency models. Our approach begins with a naïve baseline and proceeds to deconstruct the training process, isolating and examining the impact of key modules: time step discretization, time preconditioning, loss weighting, time sampling strategies, the auxiliary task with variable upper limit, and distribution-level losses. This modular analysis provides a clear view of how each element contributes to the overall performance. Following this guide, we demonstrate the ability to build models that achieve both state-of-the-art results and substantially faster convergence. Notably, for the first time, consistency models trained from scratch now surpass the leading EDM diffusion model on CIFAR-10 under the same network architecture. They achieve a remarkable 1-step FID of 2.53 and a 2-step FID of 1.92.

## 1 Introduction

Diffusion models (Sohl-Dickstein et al., 2015; Ho et al., 2020; Song & Ermon, 2019; Song et al., 2020) have established themselves as the state-of-the-art for high-fidelity data synthesis, with remarkable success in domains such as image generation (Dhariwal & Nichol, 2021; Karras et al., 2022; Peebles & Xie, 2023; Rombach et al., 2022), audio synthesis (Kong et al., 2020; Huang et al., 2023; Liu et al., 2023), and beyond. They progressively add noise to data in a forward process and then learn a neural network to reverse this process, generating data from pure noise. However, this reversal typically involves solving an ordinary or stochastic differential equation (ODE/SDE), where the iterative steps lead to prohibitively slow inference times.

To mitigate this critical drawback, a significant body of research has focused on accelerating the generation process, primarily through few-step diffusion distillation or specialized training techniques (Luhman & Luhman, 2021; Salimans & Ho, 2022; Sauer et al., 2024; Wang et al., 2023; Yin et al., 2024; Luo et al., 2023; Xie et al., 2024; Salimans et al., 2025; Song et al., 2023). Among these, consistency training (Song et al., 2023) emerges as an elegant and lightweight approach. It is built by enforcing consistency between the outputs of adjacent points that lie on the same ODE solution trajectory. This principle allows for direct, one-step generation after training. Despite the simple and complete theory, the practical implementation of consistency models is fraught with challenges. The practical application reveals that the training process can be sensitive, with critical components like loss weighting and time sampling strategies being closely coupled. The existing literature provides diverse and sometimes contradictory approaches for these components, which are typically determined on an empirical basis. This makes it difficult to understand the true impact of each module and hinders systematic improvement.

In this paper, we argue that the primary bottleneck to achieving superior performance of consistency models lies not in the already mature principle, but in the nuanced, often-overlooked details of the training pipeline. We contend that these subtle design choices can lead to substantial differences in final model performance. To untangle this complexity and provide a clear path forward, we deconstruct the consistency training process into its fundamental modules. We begin by establishing a naïve baseline model under the simple and widely used flow matching interpolant (Liu et al., 2022; Lipman et al., 2022; Albergo et al., 2023). Then, in a modular fashion, we progressively add and analyze each component. Our investigation covers: the discretization schedule, time precondition-

Table 1: Comparison of design choices between naïve baseline and our modifications on CIFAR-10.

| Settings | Baseline (Section 3.1) | Our improved model (GCM) |
|---|---|---|
| Discretization function (Section 3.2) | $h(t) = \text{const}$ | $h(t) = \epsilon e^{-\mu t}$ |
| Time preconditioning (Section 3.2) | $c(t) = t$ | $c(t) = \frac{e^{\mu t} - 1}{\mu}$ |
| Loss weighting function (Section 3.3) | $w(t) = 1$ | $w(t) = t^\gamma e^{-\lambda t}$ |
| $s$ (end time) sampling (Section 3.4) | $p(s) = \delta(s - t_0)$ | CU: $p(s) = \delta(s - t_0)$ |
| | | VU: $p(s) = p_{\text{up}}(s; \nu)$ (Eq. 17) |
| $t$ (start time) sampling (Section 3.3) | $p(t) = 1$ | $p(t) = \frac{1}{t_T - s} p_{\text{sine}}\left(\frac{t - s}{t_T - s}; a, b\right)$ (Eq. 15) |
| Loss type (Section 3.5) | $\ell_2$ norm loss (Eq. 5) | Time-aware MMD with $m, \sigma$ (Eq. 45) |
| Parameters | N/A | $\epsilon = 0.005, \mu = 3.5, \gamma = 1, \lambda = 6$ |
| | | $\nu = 4, a = 1, b = 0.01$ |
| | | CU: $m = 4, \sigma = 0.05$ |
| | | VU: $m = 2, \sigma = 0.1$ |

ing, loss weighting, time sampling strategies, the auxiliary task with variable upper limit, and the distribution-level losses.

Through this systematic and decoupled analysis, we offer a clear playbook for training high-performance consistency models. We also provide a detailed comparison of our proposed training strategy against those from prior works, particularly concerning loss weighting and time sampling schemes. By applying our proposed improvements to EDM on CIFAR-10, we achieve an FID score of 2.86, using a training budget identical to that of diffusion models (exception for a single extra forward pass during training). Furthermore, by increasing the batch size to 1024, we attain a 1-step FID of 2.53 and a 2-step FID of 1.92, surpassing the original EDM's best reported score of 1.97.

## 2 PRELIMINARIES

### 2.1 DIFFUSION MODELS

Without loss of generality, we use velocity prediction (Salimans & Ho, 2022) and flow matching interpolant (Liu et al., 2022; Lipman et al., 2022; Albergo et al., 2023) to build diffusion models. Let $p_d$ be the data distribution and $\boldsymbol{x}_t = \alpha_t \boldsymbol{x} + \sigma_t \boldsymbol{z}$ denote the noised data between a data point $\boldsymbol{x} \sim p_d$ and a Gaussian sample $\boldsymbol{z} \sim \mathcal{N}(\boldsymbol{0}, \boldsymbol{I})$, where $\alpha_t$ and $\sigma_t$ are differentiable coefficients, and $t \in [t_0, t_T]$ is the noise level. A diffusion model defines an ODE

$$\frac{d\boldsymbol{x}_t}{dt} = \boldsymbol{v}(\boldsymbol{x}_t, t) \tag{1}$$

that connects the two distributions, where $\boldsymbol{v}(\boldsymbol{x}, t)$ is the velocity field of the ODE. One uses a neural network with parameter $\theta$ to approximate the velocity filed $\boldsymbol{v}$ by minimizing a regression objective

$$\mathcal{L}_{\text{Diff}}(\theta) = \mathbb{E}_{\boldsymbol{x}, \boldsymbol{z}, t}[\|\boldsymbol{v}_\theta(\boldsymbol{x}_t, t) - (\alpha'_t \boldsymbol{x} + \sigma'_t \boldsymbol{z})\|_2^2], \tag{2}$$

where $\alpha'_t$ and $\sigma'_t$ are time derivatives. For generation, one samples $\boldsymbol{z} \sim \mathcal{N}(\boldsymbol{0}, \boldsymbol{I})$ and solves the ODE numerically backwards in time, which is an iterative and slow process.

### 2.2 CONSISTENCY MODELS

In this paper, we mainly work upon discrete-time consistency training for its simplicity and lightweight nature, and will later compare the performance of both strategies. We similarly build consistency models under the flow matching interpolant and model parametrization, which has also been applied in Zhou et al. (2025); Liu & Yue (2025); Geng et al. (2025); Peng et al. (2025). We note that the original consistency models (Song et al., 2023) use a constant upper limit (CU) $s = t_0$, and CTM (Kim et al., 2023) extends it to the case with variable upper limit (VU). During the conceptual review, we always assume the variable-upper-limit case and include an upper-limit variable $s$. We parametrize the network $\boldsymbol{f}_\theta(\boldsymbol{x}_t, t, s)$ such that $\boldsymbol{x}_s$ on the same trajectory as $\boldsymbol{x}_t$ can be predicted by

$$\boldsymbol{g}_\theta(\boldsymbol{x}_t, t, s) = \boldsymbol{x}_t + (s - t)\boldsymbol{f}_\theta(\boldsymbol{x}_t, t, s). \tag{3}$$

That means, if sufficiently trained, $\boldsymbol{f}_\theta(\boldsymbol{x}_t, t, s)$ will converge to

$$\boldsymbol{f}(\boldsymbol{x}_t, t, s) = \begin{cases} \boldsymbol{v}(\boldsymbol{x}_t, t), & \text{if } t = s, \\ \dfrac{1}{s-t} \displaystyle\int_t^s \boldsymbol{v}(\boldsymbol{x}_\tau, \tau) d\tau, & \text{if } t \neq s, \end{cases} \quad (4)$$

where $\boldsymbol{f}$ is continuous at $s = t$. Consistency models construct the loss function by enforcing the model outputs of adjacent noised points on the same ODE trajectory to be the same in an inductive manner. During consistency training, one samples a noised point $\boldsymbol{x}_t = \alpha_t \boldsymbol{x} + \sigma_t \boldsymbol{z}$ and uses the estimation $\boldsymbol{u}_t = \alpha_t' \boldsymbol{x} + \sigma_t' \boldsymbol{z}$ to approximate $\boldsymbol{v}(\boldsymbol{x}_t, t)$. For a nearby time point $r = t - \Delta t$ where $\Delta t \to 0$, the state estimation is $\hat{\boldsymbol{x}}_r = \boldsymbol{x}_t + (r - t)\boldsymbol{u}_t$[1]. Using the $\ell_2$ norm loss, the consistency training loss can be written as

$$\mathcal{L}_{\text{CT}}(\theta) = \mathbb{E}_{\boldsymbol{x}, \boldsymbol{z}, t}[w(t) \frac{1}{t-s} \|\boldsymbol{x}_t + (s-t)\boldsymbol{f}_\theta(\boldsymbol{x}_t, t, s) - (\hat{\boldsymbol{x}}_r + (s-r)\boldsymbol{f}_{\theta^-}(\hat{\boldsymbol{x}}_r, r, s))\|_2] \quad (5)$$

or the continuous-time version (see Appendix D for details)

$$\mathcal{L}_{\text{CT}}^\infty(\theta) = \mathbb{E}_{\boldsymbol{x}, \boldsymbol{z}, t}[w(t) \| \boldsymbol{f}_\theta(\boldsymbol{x}_t, t, s) - (\boldsymbol{u}_t + (s-t)\frac{d}{dt}\boldsymbol{f}_{\theta^-}(\boldsymbol{x}_t, t, s))\|_2] \quad (6)$$

used in Geng et al. (2025); Peng et al. (2025). Here, $\theta^-$ is the stop-gradient version of $\theta$. The $\ell_2$ norm loss, also used in Geng et al. (2024), is a special case of the Pseudo-Huber loss (Charbonnier et al., 1997) (with $c = 0$) first introduced to consistency models by Song & Dhariwal (2023). They both normalize the derivatives with respect to $\theta$, providing more stable training similar to the adaptive normalization technique used in Lu & Song (2024); Geng et al. (2025). The factor $\frac{1}{t-s}$ ensures that the derivative with respect to $\theta$ matches that of the diffusion loss (Eq. 2) in the limit $s \to t$ (see Appendix D for details). This equivalence is important, as we will later leverage the parallel between consistency loss Eq. 5 and diffusion loss Eq. 2.

## 3 METHOD

Our methodology is to deconstruct the complex techniques into orthogonal modules. We start with a minimal baseline and incrementally introduce our improvements, with modular performance analysis shown in Table 2 and Table 3. (Results upon the well-established baseline ECT (Geng et al., 2024) can be found in Appendix H.3.) For two-step generation, we consistently use restart sampling for the CU case and pushforward sampling for the VU case (definition see Zhou et al. (2025)).

### 3.1 NAÏVE BASELINE

We construct our baseline upon the EDM framework (Karras et al., 2022). We use the OT-FM (Lipman et al., 2022) interpolant ($\alpha_t = 1 - t$, $\sigma_t = t$) and Eq. 3 as the model parametrization. For model preconditioning, we apply $c_{\text{in}}(t) = \frac{1}{\sigma_d \sqrt{\alpha_t^2 + \sigma_t^2}}$ and $c_{\text{out}}(t) = \sigma_d \sqrt{\alpha_t'^2 + \sigma_t'^2}$ to maintain that the inputs and outputs of the model have approximately unit variance. We use consistency loss with the $\ell_2$ norm metric Eq. 5 as the loss function. To decouple the main one-step generation task CU and the assistant task VU, we fix $s = t_0$ for now to study sole CU and will later release the fixing constraint. We set $t - r = \epsilon$, where $\epsilon$ is a small positive number. One can refer to Appendix G for more implementation details.

### 3.2 TIME STEP DISCRETIZATION AND TIME PRECONDITIONING

One of the challenges in discrete-time consistency training is designing an effective time discretization schedule. Existing works usually apply EDM discretization (Song et al., 2023; Song & Dhariwal, 2023; Kim et al., 2023; Geng et al., 2024) or simply use constant step size (Sun et al., 2025). IMM (Zhou et al., 2025) adopts a discretization schedule conditioned on the LogSNR (Kingma et al., 2021). We provide further details and discussion on discretization schedules in Appendix C.

---

[1]The common convention is to sample two points $\boldsymbol{x}_t$ and $\boldsymbol{x}_r$ using the interpolant, while in the sense of $\Delta t \to 0$, they are equivalent.

Table 2: Evaluation of improvements on the constant-upper-limit (CU) variant.

| Configuration | 1-step | 2-step |
|---|---|---|
| A Baseline | 6.40 | 4.28 |
| B + Discretization Function | 5.73 | 4.40 |
| C + Time Preconditioning | 5.75 | 4.31 |
| D + Loss Weighting | 3.02 | 2.24 |
| E + $t$-sampling | 2.91 | 2.20 |
| F + Time-aware MMD Loss | 2.91 | 2.19 |

Table 3: Evaluation of improvements on the variable-upper-limit (VU) variant.

| Configuration | 1-step | 2-step |
|---|---|---|
| G Baseline + $s$-sampling | 6.59 | 4.27 |
| H + Discretization Function | 5.50 | 4.10 |
| I + Time Preconditioning | 5.49 | 4.06 |
| J + Loss Weighting | 2.94 | 2.15 |
| K + $t$-sampling | 2.91 | 2.15 |
| L + Time-aware MMD Loss | 2.86 | 2.11 |

To improve performance, Song et al. (2023) and Song & Dhariwal (2023) introduce dynamic discretization curricula, where the step size shrinks as training progresses. However, optimizing such a curriculum can be resource-intensive for new datasets or tasks. For efficiency, we define the step size $h(t) = t - r$ as a function dependent solely on time $t$. In this section, we derive this formulation from theoretical intuition and also show its link to the time preconditioning $c(t)$ of the network.

**Discretization function** $h(t)$. To derive the optimal form of $h(t)$ from first principles, we begin by analyzing the minimizer of the consistency loss. Given the definition of $\boldsymbol{g}_\theta(\boldsymbol{x}_t, t, s)$ in Eq. 3, we show that (in Appendix C) the $\ell_2$ norm loss used in $\mathcal{L}_{\text{CT}}(\theta)$ (Eq. 5) is minimized when the model output $\boldsymbol{g}_\theta$ becomes the conditional *geometric median* of $\boldsymbol{g}_{\theta^-}(\hat{\boldsymbol{x}}_r, r, s)$:

$$\boldsymbol{g}_\theta(\boldsymbol{x}_t, t, s) = \text{GM}_{\hat{\boldsymbol{x}}_r}[\boldsymbol{g}_{\theta^-}(\hat{\boldsymbol{x}}_r, r, s)|\boldsymbol{x}_t]. \tag{7}$$

This contrasts with the simpler squared $\ell_2$ loss, whose minimizer is the more tractable conditional *expectation* ($\mathbb{E}$). The non-linearity of the GM operator makes a direct error analysis intractable. To gain theoretical insight, we therefore first analyze the simpler expectation case, which allows for a rigorous derivation. Our analysis focuses on minimizing the *global cumulative error*, which propagates from the noisy end of the trajectory ($t_T$) to the clean data end ($t_0$). Under the mild assumption that the local error coefficient is constant, we prove that the global error is minimized when the step size function $h(t)$ takes the form of an *exponentially decaying function*:

$$h(t) \propto e^{-\mu t}, \quad \text{where } \mu > 0. \tag{8}$$

The full proof is provided in Appendix C (Proposition 3). This focus on global error is a key departure from other approaches. For instance, minimizing the *local* error at each step leads to a bell-shaped function for $h(t)$, similar to that used in EDM (Karras et al., 2022). However, we contend that for the highly dynamic consistency training process, minimizing the final *accumulated* error is the more critical objective. While our derivation is based on the analytically tractable expectation case, we hypothesize that the core principle remains valid for the geometric median case due to the relative stability of the data manifold. We therefore adopt the exponentially decaying function as our default, a choice that is robustly outperforms various competitors in our experiments.

**Time Preconditioning** $c(t)$. A side effect of an exponentially decaying $h(t)$ is that as $t$ approaches $t_T$, the points $t$ and $t - h(t)$ become numerically very close. This proximity makes it difficult for the neural network to distinguish between them, reducing the effective time resolution in this region. As shown in Table 2, this choice of $h(t)$ improves one-step FID at the cost of two-step performance. While Zhou et al. (2025) address a similar issue by using a linear $c(t)$ with a large time scale ($c(t) = 1000t$), that approach significantly slows training convergence. To resolve this trade-off, we introduce a time preconditioning function $c(t)$ that remaps the time input to achieve uniform resolution. This condition is met when the perceived distance $c(t) - c(t - h(t))$ between steps is constant, which implies $c'(t)h(t)$ is constant in the continuous limit where $h(t) \to 0$, yielding the general form

$$c(t) = K \int \frac{dt}{h(t)} + C, \tag{9}$$

where $K$ and $C$ are constants. With our chosen $h(t) = \epsilon e^{-\mu t}$, the resulting conditioning function is $c(t) = \frac{e^{\mu t} - 1}{\mu}$, where the constants are selected to ensure that $c(t)$ behaves like the previous $c(t) = t$ near the origin, satisfying the condition $\lim_{t \to 0} c(t)/t = 1$. As Table 2 indicates, it achieves a slightly better balance between one-step and two-step performance.

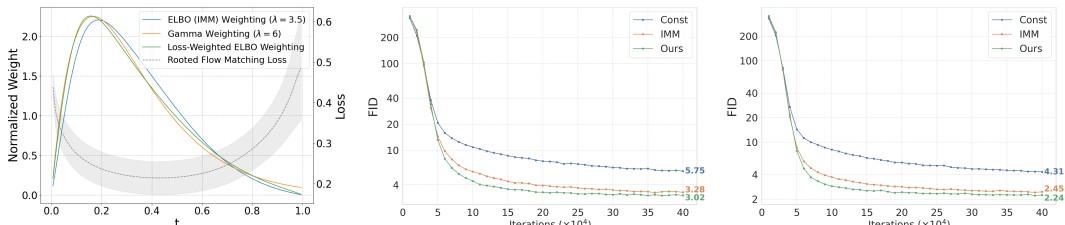

Figure 1: (**left**) The ELBO loss weighting function $w_{\text{DM}}(t)$ used by IMM (Zhou et al., 2025), the square-rooted flow matching loss $\|E_{\text{DM}}\|_2$, the loss-weighted ELBO loss weighting function $\|E_{\text{DM}}\|_2 w_{\text{DM}}(t)$, and our Gamma weighting function. (**middle**) 1-step and (**right**) 2-step FID score with respect to different loss weighting functions.

### 3.3 Loss Weighting and Time Sampling

Previous works choose loss weighting function $w(t)$ and time sampling probability density function (PDF) $p(t)$ based on empirical trials. And they often first tune the PDF $p(t)$ and then the weight $w(t)$, which makes the two parts entangled. Inspecting the loss function Eq. 5, we have

$$\mathcal{L}_{\text{CT}}(\theta) = \mathbb{E}_{\boldsymbol{x},\boldsymbol{z}}\Big[\int_{t_0}^{t_T} w(t)p(t)\|\frac{1}{t-s}(\boldsymbol{x}_t + (s-t)\boldsymbol{f}_\theta(\boldsymbol{x}_t,t,s) - (\hat{\boldsymbol{x}}_r + (s-r)\boldsymbol{f}_{\theta^-}(\hat{\boldsymbol{x}}_r,r,s)))\|_2 dt\Big],$$
(10)

where the term $w(t)p(t)$ acts as the effective weight for the error at time $t$. While the sole purpose of $w(t)$ is to modulate the errors at different times, $p(t)$ not only contributes to this weighting but also allocates computational resources. Therefore, it is more principled to first uniformly sample $t$ and determine the optimal error weighting $w(t)$ and then, adjust the sampling distribution $p(t)$ to focus resources where they are most needed while keeping the effective weight by modifying the weight to $w(t)/p(t)$.

**Loss weighting** $w(t)$. While consistency models lack a standard practice for loss weighting, the topic is well-studied in the diffusion model literature. Following Zhou et al. (2025), we draw inspiration from ELBO-based weighting schemes (Kingma & Gao, 2023). However, a direct translation of weighting functions is suboptimal because the diffusion loss does not converge to zero, making the loss value an unreliable proxy for gradient magnitude. Our approach is to instead align *the effective gradient magnitudes* of the two models. The gradient of the diffusion loss (Eq. 2) can be expressed as:

$$\nabla_\theta \mathcal{L}_{\text{Diff}}(\theta) = \mathbb{E}_{\boldsymbol{x},\boldsymbol{z},t}[2w_{\text{DM}}(t)E_{\text{DM}}\nabla_\theta \boldsymbol{v}_\theta(\boldsymbol{x}_t,t)]$$
$$= \mathbb{E}_{\boldsymbol{x},\boldsymbol{z},t}[2w_{\text{DM}}(t)\|E_{\text{DM}}\|_2 \frac{E_{\text{DM}}}{\|E_{\text{DM}}\|_2}\nabla_\theta \boldsymbol{v}_\theta(\boldsymbol{x}_t,t)],$$
(11)

where $E_{\text{DM}}$ is the error term and the fraction represents its normalized direction. Similarly, the consistency loss gradient (Eq. 5) is

$$\nabla_\theta \mathcal{L}_{\text{CM}}(\theta) = \mathbb{E}_{\boldsymbol{x},\boldsymbol{z},t}[-w_{\text{CM}}(t)\frac{E_{\text{CM}}}{\|E_{\text{CM}}\|_2}\nabla_\theta \boldsymbol{f}_\theta(\boldsymbol{x}_t,t,s)].$$
(12)

To match the gradient magnitudes, an optimal diffusion weight $w_{\text{DM}}(t)$ should be adapted for consistency models as follows:

$$w_{\text{CM}}(t) = \|E_{\text{DM}}\|_2 w_{\text{DM}}(t).$$
(13)

Since $\|E_{\text{DM}}\|_2$ is intractable during training, we estimate it by pre-training a diffusion model with the same architecture and an ELBO-based weight (Kingma & Gao, 2023; Zhou et al., 2025), then measuring its terminal root-mean-square error at each time step. We plot the ELBO weighting $w_{\text{DM}}(t)$ used in IMM (Zhou et al., 2025), the square-rooted flow matching loss $\|E_{\text{DM}}\|_2$ and the (flow matching) loss-weighted ELBO weighting $\|E_{\text{DM}}\|_2 w_{\text{DM}}(t)$ in Fig. 1 (left). We find that an instance of Gamma PDF ($p(t) \propto te^{-6t}$) provides a good approximation to $\|E_{\text{DM}}\|_2 w_{\text{DM}}(t)$, and we therefore set our default loss weighting function which we term the Gamma weight as

$$w_{\text{CM}}(t) = t^\gamma e^{-\lambda t}.$$
(14)

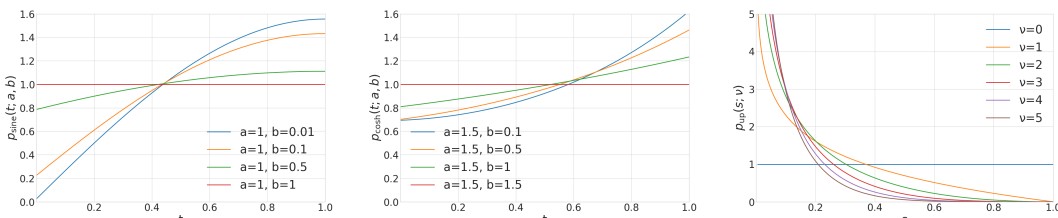

Figure 2: The probability density functions of sine distribution (left), cosh distribution (middle), and uniform-power distribution (right).

We empirically find that the shape of this weighting function near $t = t_T$ (the high-noise region) is particularly crucial for model performance. More discussions on loss weighting functions can be referred to Appendix D.3.

**Time sampling** $p(t)$. Since our ultimate goal is high-quality one-step sampling, which corresponds the model input at $t_T$, the accuracy at times closer to $t_T$ is arguably more critical. We therefore aim to allocate more training resources to this high-noise region by using importance sampling for the time variable $t$. To this end, we require the time sampling distributions to be i) positive on the training interval since $p(t)$ will be the denominator to keep the effective weight and ii) simple enough to allow for efficient inverse transform sampling. We introduce two flexible PDFs for $p(t)$. The first, which we term the *sine distribution*, is defined as:

$$p_{\text{sine}}(t; a, b) = \frac{\pi(a - b)}{2(\cos(\frac{\pi}{2}b) - \cos(\frac{\pi}{2}a))} \sin(\frac{\pi}{2}(at + b(1 - t))). \tag{15}$$

The second is the *cosh distribution*:

$$p_{\text{cosh}}(t; a, b) = \frac{a - b}{\sinh(a) - \sinh(b)} \cosh(at + b(1 - t)), \tag{16}$$

where $a \neq b$ are parameters that control the shape of each distribution, and the sampling details can be found at Appendix E. As illustrated in Fig. 2, the primary difference between these distributions lies in their convexity.

### 3.4 Auxiliary Variable-Upper-Limit Integration Task

In original consistency models (Song et al., 2023), the upper integration limit $s$ is fixed at $t_0$. While functional, this means the training target is always model-dependent, which can lead to instability. In contrast, the diffusion loss (Eq. 2) involves a data-only target, $u_t$, which contributes to its training stability. To leverage the parallel with the stable targets in diffusion models, we frame variable-upper-limit integration as an auxiliary task to stabilize training. It has also been studied in Peng et al. (2025) while our strategy differs. By allowing $s$ to vary, the stable target $u_t$ is introduced into the training process when $s = t$. Another benefit of VU is the introduction of pushforward sampling (Zhou et al., 2025) that strictly follows the ODE trajectory. This contrasts with the stochastic restart sampling used in CU models and empirically yields better performance for multi-step ($> 1$) generation (Zhou et al., 2025). Our sampling procedure for the auxiliary time point $s$ follows a two-level structure. We first sample $s$ from $[t_0, t_T]$ and then sample $t$ from the conditional range $[s, t_T]$. This process is designed to recover the standard consistency model case when $s = t_0$, allowing us to reuse established practices (see Appendix E for details). Since this is an auxiliary task, we sample $s$ from a distribution heavily skewed towards $t_0$, ensuring the standard objective remains dominant. Specifically, we use what we term the *uniform-power distribution*:

$$p_{\text{up}}(s; \nu) = \nu \int_s^1 \frac{1}{\tau}(1 - \frac{s}{\tau})^{\nu-1} d\tau. \tag{17}$$

This PDF has a singularity at $s = t_0$ and decays rapidly for $s > t_0$. The decay rate is controlled by the hyperparameter $\nu$, as illustrated in Fig. 2. Sampling from this distribution is detailed in the Appendix E. As shown in Table 3, the performance advantage of sine sampling for $t$ diminishes when the auxiliary task is introduced. We hypothesize that this is because conditioning on $s$ inherently biases the sampling of $t$ towards $t_T$, an effect that emulates the behavior of sine sampling.

Table 4: Unconditional image generation results on CIFAR-10.

| Method | FID↓ | NFE↓ | Method | FID↓ | NFE↓ |
|---|---|---|---|---|---|
| **Diffusion & Fast Samplers** | | | **Joint Training** | | |
| DDPM (Ho et al., 2020) | 3.17 | 1000 | Diff-Instruct (Luo et al., 2023) | 4.53 | 1 |
| Score SDE (deep) (Song et al., 2020) | 2.20 | 2000 | DMD (Yin et al., 2024) | 3.77 | 1 |
| EDM (Karras et al., 2022) | 1.97 | 35 | CTM (Kim et al., 2023) | 1.87 | 2 |
| Flow Matching (Lipman et al., 2022) | 6.35 | 142 | SiD (Zhou et al., 2024b) | 1.92 | 1 |
| Rectified Flow (Liu et al., 2022) | 2.58 | 127 | SiD$^2$A (Zhou et al., 2024a) | **1.50** | 1 |
| DPM-Solver (Lu et al., 2022a) | 4.70 | 10 | SiM (Luo et al., 2024) | 2.06 | 1 |
| DPM-Solver++ (Lu et al., 2022b) | 2.91 | 10 | | | |
| DPM-Solver-v3 (Zheng et al., 2023b) | 2.51 | 10 | **Few-step Training/Finetuning** | | |
| **Few-step Distillation** | | | iCT (Song & Dhariwal, 2023) | 2.83 | 1 |
| | | | | 2.46 | 2 |
| KD (Luhman & Luhman, 2021) | 9.36 | 1 | ECT (Geng et al., 2024) | 3.60 | 1 |
| PD (Salimans & Ho, 2022) | 4.51 | 2 | | 2.11 | 2 |
| DFNO (Zheng et al., 2023a) | 3.78 | 1 | sCT (Lu & Song, 2024) | 2.85 | 1 |
| 2-Rectified Flow (Liu et al., 2022) | 4.85 | 1 | | 2.06 | 2 |
| TRACT (Berthelot et al., 2023) | 3.32 | 2 | IMM (Zhou et al., 2025) | 3.20 | 1 |
| PID (Tee et al., 2024) | 3.92 | 1 | | 1.98 | 2 |
| CD (Song et al., 2023) | 2.93 | 2 | MeanFlow (Geng et al., 2025) | 2.92 | 1 |
| sCD (Lu & Song, 2024) | 3.66 | 1 | **CU (Ours)** | 2.59 | 1 |
| | 2.52 | 2 | | 2.05 | 2 |
| FACM (Peng et al., 2025) | **2.69** | 1 | **VU (Ours)** | **2.53** | 1 |
| | **1.87** | 2 | | **1.92** | 2 |

## 3.5 DISTRIBUTION-LEVEL SUPERVISION

The MMD loss (Gretton et al., 2012; Li et al., 2015) adopted by Zhou et al. (2025) introduces distributional supervision, while has a significant constraint: it requires all samples within a calculation group to share the exact same time value $t$, thereby limiting the diversity of time points. We relax this constraint with a time-aware MMD loss, which uses a modified kernel to account for temporal differences between samples (see Appendix F for details).

## 4 EXPERIMENTS

### 4.1 SYSTEM-LEVEL COMPARISON

**Overall Performance.** Our final, optimized training strategy integrates all the aforementioned improvements. We evaluate this strategy on the CIFAR-10 benchmark, comparing it against state-of-the-art generative models in Table 4. Our method achieves a 1-step FID of 2.53 and a 2-step FID of 1.92, substantially outperforming previous consistency training methods. Furthermore, it surpasses competing few-step distillation techniques, with the sole exception of our 2-step result being second to that of the recent FACM (Peng et al., 2025). When compared to few-step finetuning methods, our model not only achieves significantly better performance but does so at a less computational cost. Most remarkably, this work marks the first time a consistency model trained from scratch has outperformed the highly optimized EDM diffusion model (2 steps vs. 35 steps). These results demonstrate the power of our carefully designed and decoupled training pipeline.

Table 5: Comparison on performance vs. training cost among different methods.

| Method | Batchsize | Iterations | 1-step | 2-step |
|---|---|---|---|---|
| CT | 512 | 800K | 8.70 | 5.83 |
| iCT | 512 | 400K | 3.18 | - |
| iCT | 1024 | 400K | 2.83 | 2.46 |
| IMM | 4096 | 400K | 3.20 | 1.98 |
| MeanFlow | 1024 | 800K | 2.92 | - |
| CU (Ours) | 512 | 400K | 2.91 | 2.19 |
| CU (Ours) | 1024 | 400K | 2.66 | 2.07 |
| CU (Ours) | 1024 | 800K | 2.59 | 2.05 |
| VU (Ours) | 512 | 400K | 2.86 | 2.11 |
| VU (Ours) | 1024 | 400K | 2.63 | 1.93 |
| VU (Ours) | 1024 | 800K | 2.53 | 1.92 |

**Training Compute.** We compare the training compute of our method against existing training-from-scratch approaches in Table 5. The results demonstrate that our method achieves superior

performance for a given computational budget. For instance, when using a training budget identical to that of standard diffusion models (512 batch size, 400K iterations), our method achieves a 1-step FID of 2.86. Furthermore, with a training cost that is on par with MeanFlow (Geng et al., 2025) and substantially lower than that of IMM (Zhou et al., 2025), our method reaches its peak performance: a 1-step FID of 2.53 and a 2-step FID of 1.92. Visualizations are provided in Appendix I.

**Transferring to ImageNet.** To demonstrate the transferability of our method, we conduct experiments on ImageNet-64×64 with EDM2 (Karras et al., 2024) and ImageNet-256×256 (in the latent space) with DiT (Peebles & Xie, 2023). The results are summarized in Table 6, and the detailed experimental settings can be referred to Appendix G. The additional experiments demonstrate that our method (a) successfully adapts to a fine-

Table 6: Results on ImageNet. *indicates our reproduction.

(a) Results on ImageNet-64×64 (fine-tuning).

| Model | 1-step | 2-step |
|---|---|---|
| ECT-S | 5.51 | 3.18 |
| ECT-S* | 5.83 | 3.47 |
| CU-S (Ours) | 5.25 | 3.45 |
| ECT-XL | 3.35 | 1.96 |
| ECT-XL* | 3.56 | 2.10 |
| CU-XL (Ours) | 3.44 | 1.98 |

(b) Results on ImageNet-256×256 (latent space).

| Model | Epochs | 1-step |
|---|---|---|
| MeanFlow-B/4 | 80 | 15.53 |
| MeanFlow-B/4* | 80 | 15.43 |
| VU-B/4 (Ours) | 80 | 14.27 |
| MeanFlow-XL/2 | 40 | ∼7.8 |
| MeanFlow-XL/2* | 40 | 7.75 |
| VU-XL/2 (Ours) | 40 | 7.12 |

tuning scenario on ImageNet-64×64, outperforming ECT (Geng et al., 2024), a method specifically optimized for this setup, with minimal hyperparameter tuning, and (b) effectively transfers to training from scratch on the challenging latent-space ImageNet-256×256, achieving superior results compared to the highly-tuned method MeanFlow (Geng et al., 2025) with only minor adjustments.

## 4.2 COMPONENT-WISE ANALYSIS

We conduct ablation studies to analyze each design choice and compare them with existing works.

**Discretization step size.** Our analysis begins with the discretization function $h(t)$, with visualizations and results detailed in Fig. 7a and Table 7b respectively. While keeping the integral $\int_0^1 h(t)dt$ constant across experiments, we find that the exponentially decreasing function $h(t) = \epsilon e^{-\mu t}$ yields the optimal performance, consistent with the analysis in Section 3.2. A broader observation is that decreasing functional forms consistently outperform other shapes, such as increasing, concave, or convex ones. This finding further supports our analysis which posits that minimizing the accumulated global error is critical. Consequently, a decreasing step size, moving from clean data to pure noise, proves to be the most effective strategy. Additionally, Table 7c verifies that training with our proposed time preconditioning function converges faster than that of IMM (Zhou et al., 2025); Table 7d shows that there is no need for further discretization curriculum under our strategy.

**Loss weighting and time sampling.** As shown in Tables 2 and 3, the combination between loss weighting and the time sampling contributes the most to performance. We conduct a focused study comparing our design against existing from-scratch training methods, with results in Table 7e. Our approach surpasses the second-best IMM (Zhou et al., 2025), improving the 1-step FID by 0.72 and the 2-step FID by 0.22. For an isolated analysis of the loss weighting function's impact, see Fig. 1.

**Intensity of the auxiliary task.** We evaluate the effect of the parameter $\nu$ in the uniform-power distribution, which controls the intensity of the auxiliary variable-upper-limit task. The results in Fig. 7f indicate that $\nu = 4$ achieves the best trade-off between 1-step and 2-step performance.

**Sampling the main time point $t$.** As shown in Table 7g, our empirical comparison of monotonically increasing sampling distributions reveals that the distribution's bias towards the noise region ($t \approx t_T$) is more critical than its specific shape (*e.g.*, convexity). We found that distributions prioritizing high-$t$ values tend to accelerate convergence. Specifically, the $p_{\text{sine}}$ distribution marginally outperformed $p_{\text{cosh}}$. Given this slight advantage, we select $p_{\text{sine}}$ as our default sampling strategy.

**Performance under different model architectures.** Our method demonstrates robust performance across two common diffusion network architectures DDPM++ and NSCN++. As shown in Table 7h, both architectures are effectively supported, with DDPM++ yielding better results.

**Distribution-level supervision.** We compare our proposed time-aware MMD loss against the variant from Zhou et al. (2025), which forces shared time points within each group. The results presented

Table 7: Ablation studies with EDM architecture on CIFAR-10 dataset. Each run is trained for 400K iterations with batch size of 512.

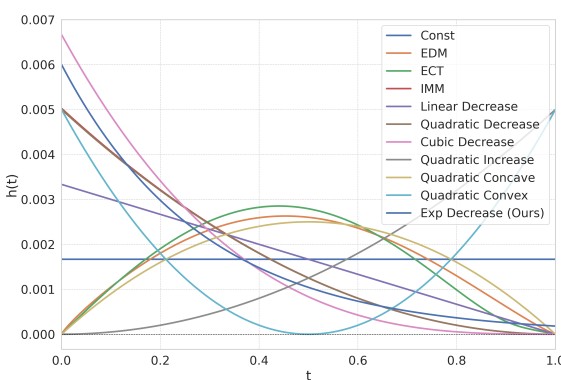

(a) Various discretization functions, normalized to a constant integral over $[0, 1]$.

(b) Performance comparison among various discretization functions.

| Discretization $h(t)$ | 1-step | 2-step |
|---|---|---|
| Constant | 4.62 | 2.50 |
| EDM's (Karras et al., 2022) | 3.29 | **2.24** |
| ECT's (Geng et al., 2024) | 3.20 | 2.27 |
| IMM's (Zhou et al., 2025) | 3.35 | 2.43 |
| Linear Decrease | 3.08 | 2.26 |
| Quadratic Decrease | 3.34 | 2.43 |
| Cubic Decrease | 3.46 | 2.65 |
| Quadratic Increase | 9.09 | 3.08 |
| Quadratic Concave | 3.38 | 2.28 |
| Quadratic Convex | 6.83 | 2.75 |
| Exponential Decrease | **3.03** | 2.26 |

(c) Time preconditioning functions.

| Conditioning | 1-step | 2-step |
|---|---|---|
| $c(t) = t$ | 5.73 | 4.42 |
| $c(t) = 1000t$ | 8.59 | 7.05 |
| $c(t) = e^{-3.5t}$ | 5.75 | 4.31 |

(d) Effect of time curriculum.

| Curriculum | 1-step | 2-step |
|---|---|---|
| ✗ | 2.91 | 2.15 |
| ✓ | 3.12 | 2.25 |

(e) Effect of collaboration between loss weighting and time sampling.

| Method | 1-step | 2-step |
|---|---|---|
| IMM | 3.63 | 2.37 |
| MeanFlow | 4.10 | 2.58 |
| FACM | 5.25 | 4.24 |
| CU (Ours) | 2.91 | 2.20 |
| VU (Ours) | 2.91 | 2.15 |

(f) Intensity of auxiliary task.

(g) Comparison on strategies for sampling the main time point $t$.

| Sampling PDF | 1-step | 2-step |
|---|---|---|
| Uniform | 3.02 | 2.24 |
| $p_{\text{sine}}(t; 1, 0.1)$ | 2.94 | 2.21 |
| $p_{\text{sine}}(t; 1, 0.01)$ | 2.91 | 2.20 |
| $p_{\text{sine}}(t; 1, 0.001)$ | 2.92 | 2.20 |
| $p_{\text{cosh}}(t; 1.5, 0)$ | 2.93 | 2.23 |

(h) Comparison on different model architectures.

| Network | 1-step | 2-step |
|---|---|---|
| DDPM++ | 2.86 | 2.11 |
| NSCN++ | 2.94 | 2.22 |

(i) Comparison on the application of MMD loss.

| Group Size | Share Time | Kernel $\sigma$ | 1-step | 2-step |
|---|---|---|---|---|
| 1 (w/o MMD) | - | - | 2.91 | 2.15 |
| 2 | ✓ | $\infty$ | 2.91 | 2.16 |
| 2 | ✗ | 0.1 | 2.86 | 2.11 |

(j) FID, training speed (sec/KIMG) and memory (G) of discrete- and continuous-time consistency losses.

| JVP Computation | 1-step | 2-step | Speed | Memory |
|---|---|---|---|---|
| Analytic | 2.93 | 2.18 | 0.82 | 38.57 |
| Finite Difference | 2.91 | 2.15 | 0.44 | 9.08 |

in Table 7i demonstrate that with a smaller batch size of 512, our approach not only improves upon the baseline but also outperforms the method from Zhou et al. (2025).

**Continuous-time vs. discrete-time derivatives.** Finally, we study the performance trade-offs between continuous-time and discrete-time consistency models within our framework. The former calculates the Jacobian-vector product (JVP) using its analytic form (`torch.jvp`), while the latter employs finite differences. As detailed in Table 7j, we observe the finite difference method achieves slightly better performance under our optimal setting, and the analytic forward-mode AD incurs significantly higher training latency and GPU memory consumption compared to the finite difference approximation.

# 5 LIMITATIONS

Our work has two main limitations. First, there is still a performance gap between our 1-step and 2-step models, indicating that a trade-off between speed and quality remains. Future research could aim to improve single-step fidelity. Second, while our method excels on low-resolution benchmarks

like CIFAR-10, its performance on high-resolution datasets such as ImageNet does not yet match that of leading diffusion models. Improving the scalability of our approach is therefore a crucial direction for future work.

## 6 CONCLUSION

In this work, we present a systematic guide to training consistency models. By moving away from entangled, empirically-driven training pipelines and adopting a modular, first-principles approach, we isolate and optimize the key components of the training process. We show that seemingly minor details, such as the discretization of the time step, loss weighting function, and sampling distributions, have a profound impact on model performance. Our final model, which integrates these improvements, sets a new state-of-the-art for consistency models trained from scratch, demonstrating that the primary obstacle to their performance is not a limitation of the underlying theory, but the complexity of the practical implementation. We believe this decoupled playbook provides a clearer and more effective path for future research in fast generative modeling.

### ETHICS STATEMENT

This research is foundational and focuses on improving the training methodology of generative models. All experiments were conducted on standard, publicly available academic datasets (e.g., CIFAR-10, ImageNet) that do not contain sensitive or personally identifiable information. Our work does not introduce new risks beyond those inherent to existing generative technologies. We support the responsible development of AI and encourage ongoing dialogue on its societal impact.

### REPRODUCIBILITY STATEMENT

To ensure our work is reproducible, we will release our source code and pretrained model weights upon publication. Our paper and its appendix provide a detailed account of all model architectures, hyperparameters, and experimental settings. The research relies on standard, publicly available datasets and common deep learning libraries, allowing the community to verify our results and build upon our findings.

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

APPENDIX

# A LLM USAGE STATEMENT

We utilized a large language model (LLM), specifically Gemini Pro 2.5, to assist with improving the clarity, grammar, and readability of our manuscript. The core ideas, experimental design, results, and conclusions presented in this paper are entirely our own.

# B RELATED WORK

## B.1 DIFFUSION MODELS, FAST SAMPLERS AND DIFFUSION DISTILLATION

Diffusion Models (Sohl-Dickstein et al., 2015; Ho et al., 2020; Song & Ermon, 2019; Song et al., 2020) and Flow Matching (Lipman et al., 2022; Liu et al., 2022) are generative frameworks trained to reverse a noise-corruption process. This is typically achieved by learning a time-dependent score or velocity function that guides noisy data back towards the original distribution. These approaches have been unified and generalized by the framework of Stochastic Interpolants (Albergo et al., 2023), which constructs a stochastic path between two data distributions. A fundamental challenge across all these models is their inference speed, as generation requires the iterative numerical solution of a reverse-time ODE or SDE, a computationally intensive process. Accelerating the iterative sampling process of diffusion models is a central challenge, primarily addressed through two distinct strategies: advanced numerical sampling and model distillation. The former focuses on developing more efficient samplers that reduce steps without altering the original model. This family comprises both training-free solvers, like high-order (Karras et al., 2022) and exponential methods (Lu et al., 2022a;b; Zheng et al., 2023b), and learnable approaches (Bao et al., 2022; Dockhorn et al., 2022; Zhou et al., 2024c) that dynamically tune sampling hyperparameters. Despite their convenience, these fast samplers typically struggle to maintain high fidelity in the ultra-low-step regime, often showing significant quality degradation below ten steps. The other stream, model distillation aims to compress a slow, multi-step teacher model into a student capable of few-step generation. This category includes a diverse range of techniques, such as progressive distillation (Salimans & Ho, 2022; Meng et al., 2023), which merges sampling steps, adversarial methods (Sauer et al., 2024; Wang et al., 2022) that leverage GAN-style (Goodfellow et al., 2014) losses, and consistency distillation (Song et al., 2023; Kim et al., 2023; Lu & Song, 2024) that directly solve the underlying probability flow ODE. While powerful, these approaches often require substantial computational resources for retraining and can introduce training instabilities or visual artifacts.

## B.2 CONSISTENCY MODELS

Consistency Models (CMs) (Song et al., 2023) represent a class of generative models capable of one-step generation while sharing the same underlying ODE trajectory as diffusion models. Their core principle is the *consistency property*, which enforces that the model's output remains identical for any point along a single ODE trajectory, achieved via a specially designed loss function. The original work proposed two distinct paradigms: Consistency Distillation (CD), which distills a pretrained diffusion model, and Consistency Training (CT), which trains a model directly from data. Subsequent research has introduced numerous refinements, such as modifying the training objective (CTM (Kim et al., 2023)), incorporating practical training enhancements (iCT (Song & Dhariwal, 2023)), leveraging pretrained models for initialization (ECM (Geng et al., 2024)), simplifying and stabilizing the framework for scalability (sCM (Lu & Song, 2024)), introducing distribution-level losses to alleviate instability (IMM (Zhou et al., 2025)), add additional diffusion supervision to stabilize training (FACM (Peng et al., 2025)). Continuous-time formulations (Lu & Song, 2024; Geng et al., 2025; Peng et al., 2025) of CMs have also gained popularity over their discrete counterparts. While CD initially showed superior performance, it inherits the limitations and computational overhead of its teacher model. CT, on the other hand, offers a more direct and independent training paradigm, but its potential has long been hindered by complex and unstable training dynamics. This work aims to study these problems and provide a practical playbook for robust and effective training.

## C   TIME DISCRETIZATION

### C.1   OUR INTUITIONS

We first analyze the simpler case where we use an $\ell_2$ loss to construct the consistency loss

$$\mathcal{L}_{\mathrm{CT}}^2(\theta) = \mathbb{E}_{\boldsymbol{x},\boldsymbol{z},t}[\|\boldsymbol{g}_\theta(\boldsymbol{x}_t,t,s) - \boldsymbol{g}_{\theta^-}(\hat{\boldsymbol{x}}_r,r,s)\|_2^2], \tag{18}$$

which is initially applied in Song et al. (2023).

For a better view of the loss minimizer, we have the following proposition.

**Proposition 1.** *When*

$$\mathcal{L}_{CT}^2(\theta) = \mathbb{E}_{\boldsymbol{x},\boldsymbol{z},t}[\|\boldsymbol{g}_\theta(\boldsymbol{x}_t,t,s) - \boldsymbol{g}_{\theta^-}(\hat{\boldsymbol{x}}_r,r,s)\|_2^2]$$

*reaches its minimum, $\boldsymbol{g}_\theta$ satisfies*

$$\boldsymbol{g}_\theta(\boldsymbol{x}_t,t,s) = \mathbb{E}_{\hat{\boldsymbol{x}}_r}[\boldsymbol{g}_{\theta^-}(\hat{\boldsymbol{x}}_r,r,s)|\boldsymbol{x}_t].$$

*Proof.* Since

$$\begin{aligned}
\mathcal{L}_{\mathrm{CT}}^2(\theta) &= \mathbb{E}_{\boldsymbol{x}_0,\boldsymbol{z},t}\|\boldsymbol{g}_\theta(\boldsymbol{x}_t,t,s) - \boldsymbol{g}_{\theta^-}(\boldsymbol{x}_r,r,s)\|_2^2 \\
&= \mathbb{E}_{\boldsymbol{x}_t,t}\mathbb{E}_{\boldsymbol{x}_r}[\mathbb{E}_{\boldsymbol{x}_0,\boldsymbol{z}}[\|\boldsymbol{g}_\theta(\boldsymbol{x}_t,t,s) - \boldsymbol{g}_{\theta^-}(\boldsymbol{x}_r,r,s)\|_2^2|\boldsymbol{x}_t,\boldsymbol{x}_r]|\boldsymbol{x}_t] \\
&= \mathbb{E}_{\boldsymbol{x}_t,t}\mathbb{E}_{\boldsymbol{x}_r}[\|\boldsymbol{g}_\theta(\boldsymbol{x}_t,t,s) - \boldsymbol{g}_{\theta^-}(\boldsymbol{x}_r,r,s)\|_2^2|\boldsymbol{x}_t] \\
&= \mathbb{E}_{\boldsymbol{x}_t,t}[\|\boldsymbol{g}_\theta(\boldsymbol{x}_t,t,s)\|_2^2 + \langle\boldsymbol{g}_\theta(\boldsymbol{x}_t,t,s),\mathbb{E}_{\boldsymbol{x}_r}[\boldsymbol{g}_{\theta^-}(\boldsymbol{x}_r,r,s)|\boldsymbol{x}_t]\rangle \\
&\quad + \mathbb{E}_{\boldsymbol{x}_r}[\|\boldsymbol{g}_{\theta^-}(\boldsymbol{x}_r,r,s)\|_2^2|\boldsymbol{x}_t]] \\
&= \mathbb{E}_{\boldsymbol{x}_t,t}[\|\boldsymbol{g}_\theta(\boldsymbol{x}_t,t,s)\|_2^2 + \langle\boldsymbol{g}_\theta(\boldsymbol{x}_t,t,s),\mathbb{E}_{\boldsymbol{x}_r}[\boldsymbol{g}_{\theta^-}(\boldsymbol{x}_r,r,s)|\boldsymbol{x}_t]\rangle \\
&\quad + \|\mathbb{E}_{\boldsymbol{x}_r}[\boldsymbol{g}_{\theta^-}(\boldsymbol{x}_r,r,s)|\boldsymbol{x}_t]\|_2^2 - \|\mathbb{E}_{\boldsymbol{x}_r}[\boldsymbol{g}_{\theta^-}(\boldsymbol{x}_r,r,s)|\boldsymbol{x}_t]\|_2^2 + \mathbb{E}_{\boldsymbol{x}_r}[\|\boldsymbol{g}_{\theta^-}(\boldsymbol{x}_r,r,s)\|_2^2|\boldsymbol{x}_t]] \\
&= \mathbb{E}_{\boldsymbol{x}_t,t}[\|\boldsymbol{g}_\theta(\boldsymbol{x}_t,t,s) - \mathbb{E}_{\boldsymbol{x}_r}[\boldsymbol{g}_{\theta^-}(\boldsymbol{x}_r,r,s)|\boldsymbol{x}_t]\|_2^2 \\
&\quad - \|\mathbb{E}_{\boldsymbol{x}_r}[\boldsymbol{g}_{\theta^-}(\boldsymbol{x}_r,r,s)|\boldsymbol{x}_t]\|_2^2 + \mathbb{E}_{\boldsymbol{x}_r}[\|\boldsymbol{g}_{\theta^-}(\boldsymbol{x}_r,r,s)\|_2^2|\boldsymbol{x}_t]],
\end{aligned}$$

we can see the latter two items are irrelevant to the gradient. Therefore, when $\mathcal{L}_{\mathrm{CT}}^2(\theta)$ achieves the minimum, we have

$$\boldsymbol{g}_\theta(\boldsymbol{x}_t,t,s) = \mathbb{E}_{\hat{\boldsymbol{x}}_r}[\boldsymbol{g}_{\theta^-}(\hat{\boldsymbol{x}}_r,r,s)|\boldsymbol{x}_t].$$

$\square$

Now we begin to consider the more complicated loss function Eq. 5. Similar to Proposition 1, we provide the following proposition.

**Proposition 2.** *When*

$$\mathcal{L}_{CT}(\theta) = \mathbb{E}_{\boldsymbol{x},\boldsymbol{z},t}[\|\boldsymbol{g}_\theta(\boldsymbol{x}_t,t,s) - \boldsymbol{g}_{\theta^-}(\hat{\boldsymbol{x}}_r,r,s)\|_2]$$

*reaches its minimum, $\boldsymbol{g}_\theta$ satisfies*

$$\boldsymbol{g}_\theta(\boldsymbol{x}_t,t,s) = \mathrm{GM}_{\hat{\boldsymbol{x}}_r}[\boldsymbol{g}_{\theta^-}(\hat{\boldsymbol{x}}_r,r,s)|\boldsymbol{x}_t],$$

*where* GM *represents the geometric median.*

*Proof.*

$$\begin{aligned}
\mathcal{L}_{\mathrm{CT}}(\theta) &= \mathbb{E}_{\boldsymbol{x}_0,\boldsymbol{z},t}\|\boldsymbol{g}_\theta(\boldsymbol{x}_t,t,s) - \boldsymbol{g}_{\theta^-}(\boldsymbol{x}_r,r,s)\|_2 \\
&= \mathbb{E}_{\boldsymbol{x}_t,t}\mathbb{E}_{\boldsymbol{x}_r}[\mathbb{E}_{\boldsymbol{x}_0,\boldsymbol{z}}[\|\boldsymbol{g}_\theta(\boldsymbol{x}_t,t,s) - \boldsymbol{g}_{\theta^-}(\boldsymbol{x}_r,r,s)\|_2|\boldsymbol{x}_t,\boldsymbol{x}_r]|\boldsymbol{x}_t] \\
&= \mathbb{E}_{\boldsymbol{x}_t,t}\mathbb{E}_{\boldsymbol{x}_r}[\|\boldsymbol{g}_\theta(\boldsymbol{x}_t,t,s) - \boldsymbol{g}_{\theta^-}(\boldsymbol{x}_r,r,s)\|_2|\boldsymbol{x}_t]
\end{aligned}$$

Since $\|\cdot\|_2$ is convex, we let its gradient (though it has no definition at $\boldsymbol{g}_\theta(\boldsymbol{x}_t,t,s) = \boldsymbol{g}_{\theta^-}(\hat{\boldsymbol{x}}_r,r,s)$, it does not matter since we can use subgradient)

$$\begin{aligned}
&\nabla_\theta\mathbb{E}_{\boldsymbol{x}_t,t}\mathbb{E}_{\boldsymbol{x}_r}[\|\boldsymbol{g}_\theta(\boldsymbol{x}_t,t,s) - \boldsymbol{g}_{\theta^-}(\boldsymbol{x}_r,r,s)\|_2|\boldsymbol{x}_t] \\
&= \mathbb{E}_{\boldsymbol{x}_t,t}\mathbb{E}_{\boldsymbol{x}_r}[\nabla_{\boldsymbol{g}_\theta}\|\boldsymbol{g}_\theta(\boldsymbol{x}_t,t,s) - \boldsymbol{g}_{\theta^-}(\boldsymbol{x}_r,r,s)\|_2|\boldsymbol{x}_t] \\
&= \mathbb{E}_{\boldsymbol{x}_t,t}\mathbb{E}_{\boldsymbol{x}_r}\Big[\frac{\boldsymbol{g}_\theta(\boldsymbol{x}_t,t,s) - \boldsymbol{g}_{\theta^-}(\boldsymbol{x}_r,r,s)}{\|\boldsymbol{g}_\theta(\boldsymbol{x}_t,t,s) - \boldsymbol{g}_{\theta^-}(\boldsymbol{x}_r,r,s)\|_2}|\boldsymbol{x}_t\Big]
\end{aligned}$$

be zero, *i.e.*,

$$\mathbb{E}_{\boldsymbol{x}_r}\left[\frac{\boldsymbol{g}_\theta(\boldsymbol{x}_t,t,s) - \boldsymbol{g}_{\theta-}(\boldsymbol{x}_r,r,s)}{\|\boldsymbol{g}_\theta(\boldsymbol{x}_t,t,s) - \boldsymbol{g}_{\theta-}(\boldsymbol{x}_r,r,s)\|_2}\Big|\boldsymbol{x}_t\right] = 0$$

which is the definition of geometric median. □

The geometric median (GM) operator is analytically less tractable than the expectation ($\mathbb{E}$), primarily due to its lack of linearity. Because this property complicates the error analysis considerably, we first present an analytical treatment of the expectation case, followed by an intuitive discussion of the geometric median case. Fig. 3 illustrates the iterative process with respect to expectation. Intuitively, errors will accumulate from pure noise (right) to clean data (left).

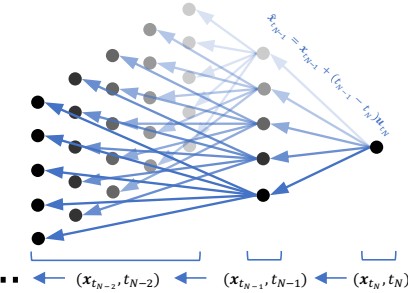

Figure 3: An illustration of consistency training process in iterative expectation. The arrows represent one Euler step approximated using $\boldsymbol{u}_t$. The model output $\boldsymbol{g}_\theta$ at each node is computed as the expectation of the multiple possible model outputs of its pointed nodes.

To investigate the global error in isolation, we have the following proposition, which introduces a specific condition ($\|\frac{d\boldsymbol{v}}{dt}\|_2 \equiv M$) for this purpose.

**Proposition 3.** *Assuming it holds that*

$$\boldsymbol{g}_\theta(\boldsymbol{x}_t,t,s) = \mathbb{E}_{\hat{\boldsymbol{x}}_r}[\boldsymbol{g}_{\theta-}(\hat{\boldsymbol{x}}_r,r,s)|\boldsymbol{x}_t].$$

*Let $\boldsymbol{v}(\boldsymbol{x}_t,t) = \mathbb{E}[\alpha'_t\boldsymbol{x} + \sigma'_t\boldsymbol{z}|\boldsymbol{x}_t]$, $t_0 < t_1 < ... < t_N = t_T$ be a partition of $[t_0,t_T]$, $h(t_n) = t_n - t_{n-1}$. We further assume that the total number of steps $N$ is fixed, and $\|\frac{d\boldsymbol{v}}{dt}\|_2 \equiv M$ where $M > 0$. In order to minimize the global error, the shape of $h(t)$ should be*

$$h(t) \propto e^{-\mu t},$$

*where $\mu$ is a positive number.*

*Proof.* For the sake of simplifying notation, let $\mathbb{E}_{\boldsymbol{x}_{t_n}|\boldsymbol{x}_{t_{n+1}}}[\cdot]$ be $\mathbb{E}_{\boldsymbol{x}_{t_n}}[\cdot|\boldsymbol{x}_{t_{n+1}}]$, $\mathbb{E}_{\boldsymbol{x}_{t_n}\leftarrow\boldsymbol{x}_{t_N}}[\cdot]$ be $\mathbb{E}_{\boldsymbol{x}_{t_{N-1}}|\boldsymbol{x}_{t_N}}\mathbb{E}_{\boldsymbol{x}_{t_{N-2}}|\boldsymbol{x}_{t_{N-1}}}...\mathbb{E}_{\boldsymbol{x}_{t_n}|\boldsymbol{x}_{t_{n+1}}}[\cdot]$, and $\hat{\boldsymbol{\mu}}_{t_n}$ be $\mathbb{E}_{\boldsymbol{x}_{t_n}\leftarrow\boldsymbol{x}_{t_N}}[\hat{\boldsymbol{x}}_{t_n}]$.

The error at step $n$ is defined by $\|e_n\|_2 = \|\boldsymbol{x}_{t_{n-1}} - \hat{\boldsymbol{\mu}}_{t_{n-1}}\|_2$. It is shown that

$$
\begin{aligned}
\|e_n\|_2 &= \|\boldsymbol{x}_{t_{n-1}} - \mathbb{E}_{\boldsymbol{x}_{t_{n-1}} \leftarrow \boldsymbol{x}_{t_N}}[\hat{\boldsymbol{x}}_{t_{n-1}}]\|_2 \\
&= \|\mathbb{E}_{\boldsymbol{x}_{t_{n-1}} \leftarrow \boldsymbol{x}_{t_N}}[\boldsymbol{x}_{t_{n-1}} - \hat{\boldsymbol{x}}_{t_{n-1}}]\|_2 \\
&= \|\mathbb{E}_{\boldsymbol{x}_{t_{n-1}} \leftarrow \boldsymbol{x}_{t_N}} \mathbb{E}_{\boldsymbol{x},\boldsymbol{z}|\boldsymbol{x}_{t_n},\boldsymbol{x}_{t_{n-1}}}[\boldsymbol{x}_{t_n} + \int_{t_n}^{t_{n-1}} \boldsymbol{v}(\boldsymbol{x}_r, r)dr - (\hat{\boldsymbol{x}}_{t_n} + (t_{n-1} - t_n)\boldsymbol{u}_{t_n})]\|_2 \\
&= \|\mathbb{E}_{\boldsymbol{x}_{t_n} \leftarrow \boldsymbol{x}_{t_N}}[\boldsymbol{x}_{t_n} + \int_{t_n}^{t_{n-1}} \boldsymbol{v}(\boldsymbol{x}_r, r)dr - (\hat{\boldsymbol{x}}_{t_n} + (t_{n-1} - t_n)\boldsymbol{v}(\hat{\boldsymbol{x}}_{t_n}, t_n))]\|_2 \\
&= \|\mathbb{E}_{\boldsymbol{x}_{t_n} \leftarrow \boldsymbol{x}_{t_N}}[\boldsymbol{x}_{t_n} - \hat{\boldsymbol{x}}_{t_n} + \int_{t_n}^{t_{n-1}} \boldsymbol{v}(\boldsymbol{x}_r, r)dr - (t_{n-1} - t_n)\boldsymbol{v}(\hat{\boldsymbol{x}}_{t_n}, t_n)]\|_2 \\
&\overset{(i)}{\leq} \|\mathbb{E}_{\boldsymbol{x}_{t_n} \leftarrow \boldsymbol{x}_{t_N}}[\boldsymbol{x}_{t_n} - \hat{\boldsymbol{x}}_{t_n}]\|_2 \\
&\quad + \|\mathbb{E}_{\boldsymbol{x}_{t_n} \leftarrow \boldsymbol{x}_{t_N}}[\int_{t_n}^{t_{n-1}} \boldsymbol{v}(\boldsymbol{x}_r, r)dr - (t_{n-1} - t_n)\boldsymbol{v}(\hat{\boldsymbol{x}}_{t_n}, t_n)]\|_2 \\
&= \|\boldsymbol{x}_{t_n} - \mathbb{E}_{\boldsymbol{x}_{t_n} \leftarrow \boldsymbol{x}_{t_N}}[\hat{\boldsymbol{x}}_{t_n}]\|_2 \\
&\quad + \|\int_{t_n}^{t_{n-1}} \boldsymbol{v}(\boldsymbol{x}_r, r)dr - (t_{n-1} - t_n)\boldsymbol{v}(\mathbb{E}_{\boldsymbol{x}_{t_n} \leftarrow \boldsymbol{x}_{t_N}}[\hat{\boldsymbol{x}}_{t_n}], t_n) \\
&\quad + (t_{n-1} - t_n)\boldsymbol{v}(\mathbb{E}_{\boldsymbol{x}_{t_n} \leftarrow \boldsymbol{x}_{t_N}}[\hat{\boldsymbol{x}}_{t_n}], t_n) - (t_{n-1} - t_n)\mathbb{E}_{\boldsymbol{x}_{t_n} \leftarrow \boldsymbol{x}_{t_N}}[\boldsymbol{v}(\hat{\boldsymbol{x}}_{t_n}, t_n)]\|_2 \\
&\overset{(ii)}{=} \|e_{n+1}\|_2 + \|(t_{n-1} - t_n)\boldsymbol{v}(\boldsymbol{x}_{t_n}, t_n) - (t_{n-1} - t_n)\boldsymbol{v}(\mathbb{E}_{\boldsymbol{x}_{t_n} \leftarrow \boldsymbol{x}_{t_N}}[\hat{\boldsymbol{x}}_{t_n}], t_n) \\
&\quad + \frac{1}{2}(t_{n-1} - t_n)^2 (\boldsymbol{v}_{\boldsymbol{x}}(\boldsymbol{x}_{t_n}, t_n)\dot{\boldsymbol{x}}_{t_n} + \boldsymbol{v}_r(\boldsymbol{x}_{t_n}, t_n)) + O((t_{n-1} - t_n)^3)\|_2 \\
&\leq \|e_{n+1}\|_2 + L(t_n - t_{n-1})\|\boldsymbol{x}_{t_n} - \mathbb{E}_{\boldsymbol{x}_{t_n} \leftarrow \boldsymbol{x}_{t_N}}[\hat{\boldsymbol{x}}_{t_n}]\|_2 \\
&\quad + \frac{1}{2}(t_{n-1} - t_n)^2 \|\boldsymbol{v}_{\boldsymbol{x}}(\boldsymbol{x}_{t_n}, t_n)\dot{\boldsymbol{x}}_{t_n} + \boldsymbol{v}_r(\boldsymbol{x}_{t_n}, t_n)\|_2 + O(t_{n-1} - t_n)^3 \\
&= (1 + Lh(t_n))\|e_{n+1}\|_2 + \frac{1}{2}(t_{n-1} - t_n)^2 \|\boldsymbol{v}_{\boldsymbol{x}}(\boldsymbol{x}_{t_n}, t_n)\dot{\boldsymbol{x}}_{t_n} + \boldsymbol{v}_r(\boldsymbol{x}_{t_n}, t_n)\|_2 \\
&\quad + O(t_{n-1} - t_n)^3,
\end{aligned}
$$

where (i) is due to the linearity of expectation and triangle inequality, and (ii) is derived by first using Taylor expansion

$$
\boldsymbol{v}(\boldsymbol{x}_r, r) = \boldsymbol{v}(\boldsymbol{x}_{t_n}, t_n) + \left.\frac{d}{d\tau}\boldsymbol{v}(\boldsymbol{x}_\tau, \tau)\right|_{\tau=t_n} (r - t_n) + O((r - t_n)^2)
$$

and

$$
\begin{aligned}
\boldsymbol{v}(\hat{\boldsymbol{x}}_{t_n}, t_n) &= \boldsymbol{v}(\hat{\boldsymbol{\mu}}_{t_n}, t_n) + \boldsymbol{v}_{\boldsymbol{x}}(\hat{\boldsymbol{\mu}}_{t_n}, t_n)(\hat{\boldsymbol{x}}_{t_n} - \hat{\boldsymbol{\mu}}_{t_n}) + \frac{1}{2}\boldsymbol{v}_{\boldsymbol{x}\boldsymbol{x}}(\hat{\boldsymbol{\mu}}_{t_n}, t_n)(\hat{\boldsymbol{x}}_{t_n} - \hat{\boldsymbol{\mu}}_{t_n})^2 \\
&\quad + O((\hat{\boldsymbol{x}}_{t_n} - \hat{\boldsymbol{\mu}}_{t_n})^3)
\end{aligned}
$$

and then we have

$$
\begin{aligned}
\int_{t_n}^{t_{n-1}} \boldsymbol{v}(\boldsymbol{x}_r, r)dr &= \int_{t_n}^{t_{n-1}} [\boldsymbol{v}(\boldsymbol{x}_{t_n}, t_n) + (\boldsymbol{v}_{\boldsymbol{x}}(\boldsymbol{x}_{t_n}, t_n)\dot{\boldsymbol{x}}_{t_n} + \boldsymbol{v}_r(\boldsymbol{x}_{t_n}, t_n))(r - t_n)]\, dr \\
&= (t_{n-1} - t_n)\boldsymbol{v}(\boldsymbol{x}_{t_n}, t_n) + \frac{1}{2}(t_{n-1} - t_n)^2 (\boldsymbol{v}_{\boldsymbol{x}}(\boldsymbol{x}_{t_n}, t_n)\dot{\boldsymbol{x}}_{t_n} + \boldsymbol{v}_r(\boldsymbol{x}_{t_n}, t_n)) \\
&\quad + O((t_{n-1} - t_n)^3)
\end{aligned}
$$

and

$$\mathbb{E}_{\boldsymbol{x}_{t_n} \leftarrow \boldsymbol{x}_{t_N}}[\boldsymbol{v}(\hat{\boldsymbol{x}}_{t_n}, t_n)] = \mathbb{E}_{\boldsymbol{x}_{t_n} \leftarrow \boldsymbol{x}_{t_N}}[\boldsymbol{v}(\hat{\boldsymbol{\mu}}_{t_n}, t_n) + \boldsymbol{v}_{\boldsymbol{x}}(\hat{\boldsymbol{\mu}}_{t_n}, t_n)(\hat{\boldsymbol{x}}_{t_n} - \hat{\boldsymbol{\mu}}_{t_n})$$

$$+ \frac{1}{2}\boldsymbol{v}_{\boldsymbol{xx}}(\hat{\boldsymbol{\mu}}_{t_n}, t_n)(\hat{\boldsymbol{x}}_{t_n} - \hat{\boldsymbol{\mu}}_{t_n})^2 + O((\hat{\boldsymbol{x}}_{t_n} - \hat{\boldsymbol{\mu}}_{t_n})^3)]$$

$$= \boldsymbol{v}(\hat{\boldsymbol{\mu}}_{t_n}, t_n) + \frac{1}{2}\boldsymbol{v}_{\boldsymbol{xx}}(\hat{\boldsymbol{\mu}}_{t_n}, t_n)\mathbb{E}_{\boldsymbol{x}_{t_n} \leftarrow \boldsymbol{x}_{t_N}}(\hat{\boldsymbol{x}}_{t_n} - \hat{\boldsymbol{\mu}}_{t_n})^2$$

$$+ O((\hat{\boldsymbol{x}}_{t_n} - \hat{\boldsymbol{\mu}}_{t_n})^3)$$

$$= \boldsymbol{v}(\hat{\boldsymbol{\mu}}_{t_n}, t_n) + \frac{1}{2}\boldsymbol{v}_{\boldsymbol{xx}}(\hat{\boldsymbol{\mu}}_{t_n}, t_n)\mathbb{E}_{\boldsymbol{x}_{t_n} \leftarrow \boldsymbol{x}_{t_N}}(\hat{\boldsymbol{x}}_{t_n} - \hat{\boldsymbol{x}}_{t_{n+1}}$$

$$- (\hat{\boldsymbol{\mu}}_{t_n} - \hat{\boldsymbol{x}}_{t_{n+1}}))^2 + O((\hat{\boldsymbol{x}}_{t_n} - \hat{\boldsymbol{\mu}}_{t_n})^3)$$

$$= \boldsymbol{v}(\hat{\boldsymbol{\mu}}_{t_n}, t_n)$$

$$+ \frac{1}{2}\boldsymbol{v}_{\boldsymbol{xx}}(\hat{\boldsymbol{\mu}}_{t_n}, t_n)\mathbb{E}_{\boldsymbol{x}_{t_{n+1}} \leftarrow \boldsymbol{x}_{t_N}}\mathbb{E}_{\boldsymbol{x}, \boldsymbol{z}|\boldsymbol{x}_{t_{n+1}}, \boldsymbol{x}_{t_n}}(t_n - t_{n+1})^2(\boldsymbol{u}_{t_{n+1}}$$

$$- \mathbb{E}_{\boldsymbol{x}_{t_{n+1}} \leftarrow \boldsymbol{x}_{t_N}}\mathbb{E}_{\boldsymbol{x}, \boldsymbol{z}|\boldsymbol{x}_{t_{n+1}}, \boldsymbol{x}_{t_n}}\boldsymbol{u}_{t_{n+1}})^2 + O((\hat{\boldsymbol{x}}_{t_n} - \hat{\boldsymbol{\mu}}_{t_n})^3)$$

$$= \boldsymbol{v}(\hat{\boldsymbol{\mu}}_{t_n}, t_n) + \frac{1}{2}(t_n - t_{n+1})^2\boldsymbol{v}_{\boldsymbol{xx}}(\hat{\boldsymbol{\mu}}_{t_n}, t_n)\text{Var}[\boldsymbol{u}_{t_{n+1}}]$$

$$+ O((t_n - t_{n+1})^3).$$

We have now arrived at the recurrence inequality

$$\|\boldsymbol{e}_n\|_2 \leq (1 + Lh(t_n))\|\boldsymbol{e}_{n+1}\|_2 + \frac{1}{2}M(t_{n-1} - t_n)^2 + O(t_{n-1} - t_n)^3. \tag{19}$$

By unrolling the recurrence, we have

$$\|\boldsymbol{e}_0\|_2 \leq \frac{1}{2}M\sum_{j=1}^{N}\left(\prod_{k=1}^{j-1}(1 + Lh(t_k))\right)h^2(t_j)$$

$$\leq \frac{1}{2}M\sum_{j=1}^{N}\left(\prod_{k=1}^{j-1}e^{Lh(t_k)}\right)h^2(t_j)$$

$$= \frac{1}{2}M\sum_{j=1}^{N}e^{t_{j-1}-t_0}h^2(t_j)$$

Considering a small time interval $dt$ near $t$, the number of steps within $dt$ is $\frac{dt}{h(t)}$. Hence, in the limit $h(t_j) \to 0$, its continuous form can be written as

$$\|\boldsymbol{e}_0\|_2 \leq \frac{1}{2}M\int_{t_0}^{t_T}e^{L(t-t_0)}h(t)dt \tag{20}$$

We consider the optimization problem: minimizing the error functional

$$J[h] = \frac{1}{2}M\int_{t_0}^{t_T}e^{L(t-t_0)}h(t)dt, \tag{21}$$

given the constraint condition on the functional of step number

$$C[h] = \int_{t_0}^{t_T}(1/h(t))dt = N. \tag{22}$$

We use Lagrangian multiplier method, and define a new functional

$$L[h] = J[h] + \lambda C[h] = \int_{t_0}^{t_T}\left(\frac{1}{2}Me^{L(t-t_0)}h(t) + \frac{\lambda}{h(t)}\right)dt.$$

Let

$$\frac{\partial F}{\partial h} = \frac{1}{2}Me^{L(t-t_0)} - \frac{\lambda}{h(t)^2} = 0,$$

we have

$$h(t) = \sqrt{\frac{2\lambda}{M}} e^{\frac{L}{2}(t_0 - t)}.$$

Therefore we have

$$h(t) \propto e^{-\mu t}$$

where $\mu = \frac{L}{2} > 0$. $\qquad\square$

*Remark* 4. In the static case, *i.e.*, with a pretrained diffusion model, we are more concerned with the local error introduced by the item $\frac{1}{2}(t_{n-1} - t_n)^2 \|\boldsymbol{v_x}(\boldsymbol{x}_{t_n}, t_n)\dot{\boldsymbol{x}}_{t_n} + \boldsymbol{v}_r(\boldsymbol{x}_{t_n}, t_n)\|_2$, since $\|\boldsymbol{v_x}(\boldsymbol{x}_{t_n}, t_n)\dot{\boldsymbol{x}}_{t_n} + \boldsymbol{v}_r(\boldsymbol{x}_{t_n}, t_n)\|_2$ usually changes a lot over time. For local error, we can assume $\|\frac{d\boldsymbol{v}}{dt}\|_2$ is bounded by some positive number, following a similar line of reasoning we have

$$h(t) \propto \frac{1}{\sqrt{\|\frac{d\boldsymbol{v}}{dt}\|_2}}.$$

We train a diffusion model and calculate $\|\frac{d\boldsymbol{v}}{dt}\|_2$ and the optimal $h(t)$ conditioned on local error in Fig. 4, which resembles the EDM discretization strategy (see Fig. 7a). However, in the context

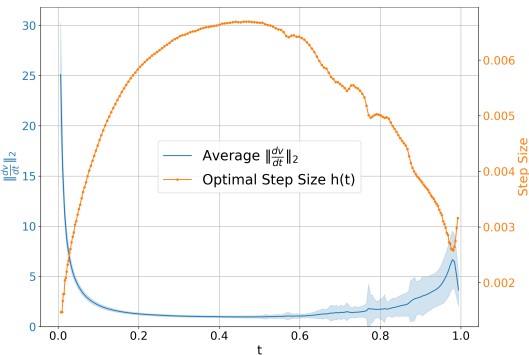

Figure 4: An illustration of the value $\|\frac{d\boldsymbol{v}}{dt}\|_2$ over time $t$ and the corresponding optimal $h(t)$ shape with respect to local error.

of consistency training, we find that due to the highly complex training dynamics, a more effective strategy is to treat the local error coefficient as a constant and focus instead on minimizing the global cumulative error.

The non-linearity of the GM operator prevents a straightforward extension of the proof in Proposition 3. Specifically, if we substitute the expectation $\mathbb{E}$ with GM, the argument breaks down at inequality (i). Also, (ii) does not hold since $\text{GM}_{\boldsymbol{x}_{t_n} \leftarrow \boldsymbol{x}_{t_N}} \boldsymbol{v_x}(\hat{\boldsymbol{\mu}}_{t_n}^{\text{G}}, t_n)(\hat{\boldsymbol{x}}_{t_n} - \hat{\boldsymbol{\mu}}_{t_n}^{\text{G}})$ where $\hat{\boldsymbol{\mu}}_{t_n}^{\text{G}} = \text{GM}_{\boldsymbol{x}_{t_n} \leftarrow \boldsymbol{x}_{t_N}} \hat{\boldsymbol{x}}_{t_n}$ is no longer guaranteed to be zero, which in turn affects the coefficient of the $(t_{n-1} - t_n)^2$ term in the recurrence relation (Eq. 19).

To proceed, we make an intuitive assumption. Given the relative stability of the data manifold, we hypothesize that inequality (i) still holds approximately. However, deriving the new coefficient for the second-order term remains analytically intractable. We therefore also treat it as a constant and continue using the function $h(t) = \epsilon e^{-\mu t}$. Empirically, we find this choice to be highly effective.

## C.2 FURTHER DETAILS ON EXISTING DISCRETIZATION STRATEGIES

**EDM discretization.** The EDM discretization follows (Eq. (269) in Karras et al. (2022))

$$\sigma_{i<N} = \left(\sigma_{\max}^{\frac{1}{\rho}} + \frac{i}{N-1}(\sigma_{\min}^{\frac{1}{\rho}} - \sigma_{\max}^{\frac{1}{\rho}})\right), \quad \sigma_N = 0. \tag{23}$$

To transform from $\sigma$ to $t$, we use the relation

$$t = \frac{2}{\pi} \arctan(\frac{\sigma}{\sigma_d}) \tag{24}$$

provided by Lu & Song (2024) (Eq. (30)), and we multiply a factor of $\frac{2}{\pi}$ since our time interval is $[0, 1]$). Hence,

$$
\begin{aligned}
h_{\text{EDM}}(t_i) &= t_i - t_{i+1} \\
&= \frac{2}{\pi} \arctan(\frac{\sigma_{i+1}}{\sigma_d}) - t_i \\
&= \frac{2}{\pi} \arctan(\frac{(\sigma_i^{\frac{1}{\rho}} - A)^\rho}{\sigma_d}) - t_i \\
&= \frac{2}{\pi} \arctan(\frac{((\sigma_d \tan(\frac{\pi}{2}t_i))^{\frac{1}{\rho}} - A)^\rho}{\sigma_d}) - t_i,
\end{aligned}
$$

where $A = \frac{1}{N-1}(\sigma_{\min}^{\frac{1}{\rho}} - \sigma_{\max}^{\frac{1}{\rho}})$, $\sigma_d = 0.5$, $\sigma_{\max} = 80$, $\sigma_{\min} = 0.002$, $\rho = 7$, $N$ is the number of discretization steps. As shown in Fig. 7a, the (translated) $h(t)$ of EDM discretization is bell-shaped.

**ECT discretization.** Following EDM (Karras et al., 2022), ECT (Geng et al., 2024) samples time points in the $\sigma$ space, which we term as $\sigma_t$ and $\sigma_r$. Different from EDM, its discretization follows

$$\sigma_r = \sigma_t(1 - \epsilon(1 + \frac{k}{1 + e^{b\sigma_t}})), \tag{25}$$

where $k = 8, b = 1$. By substituting $\sigma_t$ with $t$ into Eq. 25 according to Eq. 24, we have

$$
\begin{aligned}
h_{\text{ECT}}(t) &= t - r \\
&= t - \frac{2}{\pi} \arctan(\tan(\frac{\pi}{2}t)(1 - \epsilon(1 + \frac{k}{1 + e^{b\sigma_d \tan(\frac{\pi}{2}t)}}))).
\end{aligned}
$$

**IMM discretization.** The default setting of discretization in IMM (Zhou et al., 2025) is 'constant decrement in $\eta(t)$', where $\eta(t) = \frac{\sigma_t}{\alpha_t}$. Specifically,

$$
\begin{aligned}
h_{\text{IMM}}(t) &= t - r \\
&= t - \eta^{-1}(\eta(t) - \epsilon)) \\
&= t - \frac{\frac{t}{1-t} - \epsilon}{1 + \frac{t}{1-t} - \epsilon} \\
&= t - \frac{t - \epsilon(1-t)}{1 - \epsilon(1-t)} \\
&= \frac{\epsilon(1-t)^2}{1 - \epsilon(1-t)},
\end{aligned}
$$

where $\epsilon$ is a small positive number. The discretization function $h(t)$ used in IMM (Zhou et al., 2025) converges to a quadratic decreasing function, $\epsilon(1-t)^2$, in the limit as $\epsilon$ approaches zero. Observing that IMM employs a small value on CIFAR-10 ($\epsilon \approx 0.00488$), we adopt this limiting form as our baseline. Consequently, we set $\epsilon = 0.005$ and use $h(t) = \epsilon(1-t)^2$ as our baseline discretization function.

### C.3 DETAILS ON TESTED DESCRITIZATION FUNCTIONS

We detail each tested discretization function $h(t)$ in Section 4.2. The accuracy of the finite-difference approximation for the Jacobian-vector product (JVP) depends on the magnitude of $h(t)$; for practical reasons, this magnitude is limited by numerical precision and cannot be arbitrarily small. To ensure a fair comparison, it is therefore necessary to control for similar magnitudes. Consequently, while the optimal $h(t)$ is derived under the condition of a constant number of steps in Proposition 3, here

we impose the constraint that the integral $\int_0^t h(t)dt$ is constant. In the following, we set $h(t) = 0.005(1 - t)^2$ as as our baseline. For all other $h(t)$ functions, the value of $\epsilon$ is adjusted such that their integral is equal to that of the baseline (except for $h_{\text{EDM}}(t)$ which adjusts $A$ where $A < 0$).

**Constant function.** Constant function represents the most intuitive and straightforward discretization strategy:

$$h(t) = \epsilon \tag{26}$$

**EDM function.** EDM discretization is a high-performance strategy in diffusion models, which has been widely adopted by consistency models, such as the original consistency models (Song et al., 2023) and iCT (Song & Dhariwal, 2023).

$$h_{\text{EDM}}(t) = \frac{2}{\pi} \arctan\left(\frac{((0.5 \tan(\frac{\pi}{2}t))^{\frac{1}{7}} - A)^7}{0.5}\right) - t \tag{27}$$

**ECT function.**

$$h_{\text{ECT}}(t) = t - \frac{2}{\pi} \arctan\left(\tan\left(\frac{\pi}{2}t\right)\left(1 - \epsilon\left(1 + \frac{8}{1 + e^{0.5 \tan(\frac{\pi}{2}t)}}\right)\right)\right) \tag{28}$$

**IMM function.**

$$h_{\text{IMM}}(t) = \frac{\epsilon(1 - t)^2}{1 - \epsilon(1 - t)} \tag{29}$$

**Linear decreasing function.**

$$h(t) = \epsilon(1 - t) \tag{30}$$

**Quadratic decreasing function.**

$$h(t) = \epsilon(1 - t)^2 \tag{31}$$

**Cubic decreasing function.**

$$h(t) = \epsilon(1 - t)^3 \tag{32}$$

**Quadratic increasing function.**

$$h(t) = \epsilon t^2 \tag{33}$$

**Quadratic concave function.**

$$h(t) = \epsilon t(1 - t) \tag{34}$$

**Quadratic convex function.**

$$h(t) = \epsilon(0.5 - t)^2 \tag{35}$$

**Exponential decreasing function (Ours).**

$$h(t) = \epsilon e^{-3.5t} \tag{36}$$

We find the performance of $h(t) = 0.005e^{-3.5t}$ similar to that of $h(t) = \epsilon e^{-3.5t}$ where $\epsilon$ is tuned to match the integral of the baseline. For the sake of simplicity, we therefore set $h(t) = 0.005e^{-3.5t}$ as the default.

## C.4 DETAILED DERIVATION OF TIME PRECONDITIONING FUNCTION

The derivation for this form is based on the constraint that the function behaves like the identity near the origin, i.e., $\lim_{t \to 0} c(t)/t = 1$. This ensures that our preconditioning behaves similar to the

naïve choice $c(t) = t$. The derivation is as follows:

$$
\begin{aligned}
\lim_{t \to 0} \frac{c(t)}{t} &= \lim_{t \to 0} \frac{K \int \frac{dt}{h(t)} + C}{t} \\
&= \lim_{t \to 0} \frac{\frac{K}{\epsilon \mu} e^{\mu t} + C}{t} \\
&= \lim_{t \to 0} \frac{\frac{K}{\epsilon \mu} (1 + \mu t + O(t^2)) + C}{t} \\
&= \lim_{t \to 0} \frac{K}{\epsilon} + \frac{\frac{K}{\epsilon \mu} + C}{t} + O(t) \\
&= 1.
\end{aligned}
\tag{37}
$$

For this limit to hold, the constant term must be 1 and the $\frac{1}{t}$ term must be 0. This gives us two conditions: i) $\frac{K}{\epsilon} = 1 \implies K = \epsilon$ and ii) $\frac{K}{\epsilon \mu} + C = 0 \implies C = -\frac{K}{\epsilon \mu} = -\frac{1}{\mu}$. Substituting these back into the general form for $c(t)$ gives the correct expression: $c(t) = \frac{e^{\mu t} - 1}{\mu}$.

### C.5 Discussion on Analytic and Finite-difference JVP Calculation

As shown in Table 7 (j), the finite-difference version of JVP calculation slightly outperforms the analytic counterpart. We attribute this performance difference to the distinct practical behaviors of the analytic (continuous-time) and finite-difference (discrete-time) formulations. As noted by prior work (e.g., Fig. 1 (b) in Song & Dhariwal (2023), analysis in Lu & Song (2024)), the analytic version, which relies on 'torch.jvp' for Jacobian-vector products, is often more sensitive to implementation details (like time embeddings) and requires additional stabilization techniques. This inherent practical instability can overshadow the theoretical benefits of avoiding discretization error.

## D Details on Training Objectives and Loss Weighting

### D.1 Discrete- and Continuous-time Consistency Training Objective

Under the flow matching interpolant $\boldsymbol{x}_t = \alpha_t \boldsymbol{x} + \sigma_t \boldsymbol{z}$ and model parametrization Eq. 3, we will derive the continuous version of consistency training loss Eq. 5.

We begin by calculating the derivative of Eq. 5.

$$
\begin{aligned}
\nabla_\theta \mathcal{L}_{\mathrm{CT}}(\theta) &= \nabla_\theta \mathbb{E}_{\boldsymbol{x}, \boldsymbol{z}, t} \left[ w(t) \frac{1}{t-s} \| \boldsymbol{x}_t + (s-t) \boldsymbol{f}_\theta(\boldsymbol{x}_t, t, s) - (\hat{\boldsymbol{x}}_r + (s-r) \boldsymbol{f}_{\theta^-}(\hat{\boldsymbol{x}}_r, r, s)) \|_2 \right] \\
&= \mathbb{E}_{\boldsymbol{x}, \boldsymbol{z}, t} \Big[ -w(t) \cdot \\
&\quad \frac{\boldsymbol{x}_t + (s-t) \boldsymbol{f}_\theta(\boldsymbol{x}_t, t, s) - (\hat{\boldsymbol{x}}_r + (s-r) \boldsymbol{f}_{\theta^-}(\hat{\boldsymbol{x}}_r, r, s))}{\| \boldsymbol{x}_t + (s-t) \boldsymbol{f}_\theta(\boldsymbol{x}_t, t, s) - (\hat{\boldsymbol{x}}_r + (s-r) \boldsymbol{f}_{\theta^-}(\hat{\boldsymbol{x}}_r, r, s)) \|_2} \nabla_\theta \boldsymbol{f}_\theta(\boldsymbol{x}_t, t, s) \Big] \\
&= \mathbb{E}_{\boldsymbol{x}, \boldsymbol{z}, t} \Big[ -w(t) \frac{1}{r-t} \cdot \\
&\quad \frac{\boldsymbol{x}_t + (r-t) \boldsymbol{f}_\theta(\boldsymbol{x}_t, t, s) + (s-r) \boldsymbol{f}_{\theta^-}(\boldsymbol{x}_t, t, s) - (\hat{\boldsymbol{x}}_r + (s-r) \boldsymbol{f}_{\theta^-}(\hat{\boldsymbol{x}}_r, r, s))}{\| \boldsymbol{x}_t + (r-t) \boldsymbol{f}_\theta(\boldsymbol{x}_t, t, s) + (s-r) \boldsymbol{f}_{\theta^-}(\boldsymbol{x}_t, t, s) - (\hat{\boldsymbol{x}}_r + (s-r) \boldsymbol{f}_{\theta^-}(\hat{\boldsymbol{x}}_r, r, s)) \|_2} \cdot \\
&\quad (r-t) \nabla_\theta \boldsymbol{f}_\theta(\boldsymbol{x}_t, t, s) \Big] \\
&= \nabla_\theta \mathbb{E}_{\boldsymbol{x}, \boldsymbol{z}, t} \Big[ -w(t) \frac{1}{r-t} \| \boldsymbol{x}_t + (r-t) \boldsymbol{f}_\theta(\boldsymbol{x}_t, t, s) + (s-r) \boldsymbol{f}_{\theta^-}(\boldsymbol{x}_t, t, s) \\
&\quad - (\hat{\boldsymbol{x}}_r + (s-r) \boldsymbol{f}_{\theta^-}(\hat{\boldsymbol{x}}_r, r, s)) \|_2 \Big] \\
&= \nabla_\theta \mathbb{E}_{\boldsymbol{x}, \boldsymbol{z}, t} \Big[ -w(t) \frac{1}{r-t} \cdot \\
&\quad \| (t-r) \boldsymbol{u}_t - (t-r) \boldsymbol{f}_\theta(\boldsymbol{x}_t, t, s) + (s-r) (\boldsymbol{f}_{\theta^-}(\boldsymbol{x}_t, t, s) - \boldsymbol{f}_{\theta^-}(\hat{\boldsymbol{x}}_r, r, s)) \|_2 \Big] \\
&= \nabla_\theta \mathbb{E}_{\boldsymbol{x}, \boldsymbol{z}, t} \Big[ w(t) \| \boldsymbol{u}_t - \boldsymbol{f}_\theta(\boldsymbol{x}_t, t, s) + (s-r) \frac{\boldsymbol{f}_{\theta^-}(\boldsymbol{x}_t, t, s) - \boldsymbol{f}_{\theta^-}(\hat{\boldsymbol{x}}_r, r, s)}{t-r} \|_2 \Big].
\end{aligned}
$$

That is,

$$\nabla_\theta \mathcal{L}_{\text{CT}}(\theta) = \nabla_\theta \mathbb{E}_{\boldsymbol{x},\boldsymbol{z},t}[w(t)\|\boldsymbol{u}_t - \boldsymbol{f}_\theta(\boldsymbol{x}_t, t, s) + (s - r)\frac{\boldsymbol{f}_{\theta^-}(\boldsymbol{x}_t, t, s) - \boldsymbol{f}_{\theta^-}(\hat{\boldsymbol{x}}_r, r, s)}{t - r}\|_2]. \quad (38)$$

Hence we have

$$\nabla_\theta \mathcal{L}_{\text{CT}}^\infty(\theta) = \lim_{r \to t} \nabla_\theta \mathcal{L}_{\text{CT}}(\theta)$$

$$= \lim_{r \to t} \nabla_\theta \mathbb{E}_{\boldsymbol{x},\boldsymbol{z},t}[w(t)\|\boldsymbol{u}_t - \boldsymbol{f}_\theta(\boldsymbol{x}_t, t, s) + (s - r)\frac{\boldsymbol{f}_{\theta^-}(\boldsymbol{x}_t, t, s) - \boldsymbol{f}_{\theta^-}(\hat{\boldsymbol{x}}_r, r, s)}{t - r}\|_2$$

$$= \nabla_\theta \mathbb{E}_{\boldsymbol{x},\boldsymbol{z},t}[w(t)\|\boldsymbol{u}_t - \boldsymbol{f}_\theta(\boldsymbol{x}_t, t, s) + (s - t)\frac{d}{dt}\boldsymbol{f}_{\theta^-}(\boldsymbol{x}_t, t, s)\|_2.$$

In other words, $\mathcal{L}_{\text{CT}}^\infty(\theta)$ can be written as

$$\mathcal{L}_{\text{CT}}^\infty(\theta) = \mathbb{E}_{\boldsymbol{x},\boldsymbol{z},t}[w(t)\|\boldsymbol{u}_t - \boldsymbol{f}_\theta(\boldsymbol{x}_t, t, s) + (s - t)\frac{d}{dt}\boldsymbol{f}_{\theta^-}(\boldsymbol{x}_t, t, s)\|_2,$$

which is Eq. 6.

### D.2  THE PARALLEL BETWEEN CONSISTENCY AND DIFFUSION LOSSES

According to Eq. 38 and the fact that $s \leq r \leq t$, we have

$$\lim_{s \to t} \nabla_\theta \mathcal{L}_{\text{CT}}(\theta) = \lim_{s \to t} \nabla_\theta \mathbb{E}_{\boldsymbol{x},\boldsymbol{z},t}[w(t)\cdot$$

$$\|\boldsymbol{u}_t - \boldsymbol{f}_\theta(\boldsymbol{x}_t, t, s) + (s - r)\frac{\boldsymbol{f}_{\theta^-}(\boldsymbol{x}_t, t, s) - \boldsymbol{f}_{\theta^-}(\hat{\boldsymbol{x}}_r, r, s)}{t - r}\|_2]$$

$$= \nabla_\theta \mathbb{E}_{\boldsymbol{x},\boldsymbol{z},t}[w(t)\|\boldsymbol{u}_t - \boldsymbol{f}_\theta(\boldsymbol{x}_t, t, s)\|_2]$$

$$= \nabla_\theta \mathcal{L}_{\text{Diff}}(\theta),$$

where we ignore the difference in the term $w(t)$ between consistency loss and diffusion loss. Therefore, to leverage the established practices of diffusion models according to the parallel between consistency and diffusion loss, we use a baseline loss Eq. 5.

From the above we know that $\mathcal{L}_{\text{CT}}(\theta)$ has several equivalent variants (in the sense of equal gradient with respect to $\theta$), such as

$$\mathcal{L}_{\text{CT}}^{(1)}(\theta) = \mathbb{E}_{\boldsymbol{x},\boldsymbol{z},t}[w(t)\|\boldsymbol{u}_t - \boldsymbol{f}_\theta(\boldsymbol{x}_t, t, s) + (s - r)\frac{\boldsymbol{f}_{\theta^-}(\boldsymbol{x}_t, t, s) - \boldsymbol{f}_{\theta^-}(\hat{\boldsymbol{x}}_r, r, s)}{t - r}\|_2] \quad (39)$$

and

$$\mathcal{L}_{\text{CT}}^{(2)}(\theta) = \mathbb{E}_{\boldsymbol{x},\boldsymbol{z},t}[w(t)\frac{1}{t - r}\cdot$$

$$\|\boldsymbol{x}_t + (r - t)\boldsymbol{f}_\theta(\boldsymbol{x}_t, t, s) + (s - r)\boldsymbol{f}_{\theta^-}(\boldsymbol{x}_t, t, s) - (\hat{\boldsymbol{x}}_r + (s - r)\boldsymbol{f}_{\theta^-}(\hat{\boldsymbol{x}}_r, r, s))\|_2].$$
$$(40)$$

### D.3  MORE DISCUSSIONS ON LOSS WEIGHTING

Although consistency loss Eq. 5 appears different to diffusion loss Eq. 2, consistency loss aims to learn the integral of diffusion models. During inference, a diffusion model numerically integrates the instantaneous velocity field $\mathbb{E}[\boldsymbol{z} - \boldsymbol{x}|\boldsymbol{x}_t]$ (taking the OT-FM interpolant as example) to generate a sample. The core objective of consistency model training is to learn the result of this exact same integration process. Given that both diffusion model inference and consistency training are fundamentally concerned with integrating the same velocity field, we hypothesize that an effective loss

weighting scheme designed to modulate errors across the integration path for a diffusion model is directly applicable to training a consistency model.

Our goal is not to equate the full gradient norms, but to align their dominant scalar components. We observe that while the parameter gradient ($\nabla_\theta f_\theta$) tends to be stable (otherwise the training is easy to collapse), the error term ($E_{\mathrm{CM}}$) can be highly volatile. Prior methods (e.g., Pseudo-Huber loss in Song & Dhariwal (2023), adaptive weighting in Lu & Song (2024)) stabilize it by normalizing this error term, making the loss term $\frac{E_{\mathrm{CM}}}{\|E_{\mathrm{CM}}\|_2}\nabla_\theta f_\theta$ (and so do $\frac{E_{\mathrm{DM}}}{\|E_{\mathrm{DM}}\|_2}\nabla_\theta v_\theta$) have a roughly uniform magnitude across time. This means that the overall gradient magnitude is now dominated by the external scalar weighting function. We can therefore transfer the benefits of a high-performance DM strategy by aligning these dominant scalar functions. We achieve this by setting our CM weight $w_{\mathrm{CM}}(t) \propto \|E_{\mathrm{DM}}\|_2 w_{\mathrm{DM}}(t)$, ensuring our weighting's temporal distribution matches that of a ELBO-weighted DM.

## E  DETAILS ON SAMPLING TIME POINTS

### E.1  SAMPLING FROM SINE DISTRIBUTION AND COSH DISTRIBUTION

We use inverse transform sampling to sample from sine distribution (Eq. 15) and cosh distribution (Eq. 16). For simplicity, we assume the sampling interval is $[0, 1]$.

**Sine distribution** with PDF

$$p_{\mathrm{sine}}(t; a, b) = \frac{\frac{\pi}{2}(a-b)}{\cos(\frac{\pi}{2}b) - \cos(\frac{\pi}{2}a)} \sin\left(\frac{\pi}{2}(at + b(1-t))\right).$$

We first calculate its cumulative distribution function (CDF) $F_{\mathrm{sine}}(t) = \int_0^t p_{\mathrm{sine}}(\tau)d\tau$.

$$
\begin{aligned}
F_{\mathrm{sine}}(t) &= \int_0^t p_{\mathrm{sine}}(\tau; a, b)d\tau \\
&= \frac{\frac{\pi}{2}(a-b)}{\cos(\frac{\pi}{2}b) - \cos(\frac{\pi}{2}a)} \int_0^t \sin(\frac{\pi}{2}(a-b)\tau + \frac{\pi}{2}b)d\tau \\
&= \frac{\frac{\pi}{2}(a-b)}{\cos(\frac{\pi}{2}b) - \cos(\frac{\pi}{2}a)} \left[ -\frac{2}{\pi(a-b)}\cos(\frac{\pi}{2}(a-b)\tau + \frac{\pi}{2}b) \right]_0^t \\
&= -\frac{1}{\cos(\frac{\pi}{2}b) - \cos(\frac{\pi}{2}a)} \left( \cos(\frac{\pi}{2}(a-b)t + \frac{\pi}{2}b) - \cos(\frac{\pi}{2}b) \right) \\
&= \frac{1}{\cos(\frac{\pi}{2}b) - \cos(\frac{\pi}{2}a)} \left( \cos(\frac{\pi}{2}b) - \cos(\frac{\pi}{2}((a-b)t + b)) \right).
\end{aligned}
$$

Let $F_{\mathrm{sine}}(t) = u$, where $u \sim \mathcal{U}(0, 1)$, we have

$$
\begin{aligned}
u &= \frac{1}{\cos(\frac{\pi}{2}b) - \cos(\frac{\pi}{2}a)} \left( \cos(\frac{\pi}{2}b) - \cos(\frac{\pi}{2}((a-b)t + b)) \right) \\
u\left(\cos(\frac{\pi}{2}b) - \cos(\frac{\pi}{2}a)\right) &= \cos(\frac{\pi}{2}b) - \cos(\frac{\pi}{2}((a-b)t + b)) \\
\cos(\frac{\pi}{2}((a-b)t + b)) &= \cos(\frac{\pi}{2}b) - u\left(\cos(\frac{\pi}{2}b) - \cos(\frac{\pi}{2}a)\right) \\
\frac{\pi}{2}((a-b)t + b) &= \arccos\left(\cos(c) - u\left(\cos(\frac{\pi}{2}b) - \cos(\frac{\pi}{2}a)\right)\right).
\end{aligned}
$$

Hence

$$t = \frac{2}{\pi(a-b)}\left[ \arccos\left( \cos\left(\frac{\pi}{2}b\right) - u\left(\cos\left(\frac{\pi}{2}b\right) - \cos\left(\frac{\pi}{2}a\right)\right)\right) - \frac{\pi}{2}b \right]. \tag{41}$$

Therefore, we can first sample $u \sim \mathcal{U}(0, 1)$ and then get $t$ using Eq. 41.

**Cosh distribution** with PDF

$$p_{\text{cosh}}(t; a, b) = \frac{a - b}{\sinh(a) - \sinh(b)} \cosh(at + b(1 - t)).$$

We similarly begin to calculate the CDF.

$$F_{\text{cosh}}(t) = \int_0^t p_{\text{cosh}}(\tau; a, b) d\tau$$

$$= \frac{a - b}{\sinh(a) - \sinh(b)} \int_0^t \cosh((a - b)\tau + b) d\tau$$

$$= \frac{a - b}{\sinh(a) - \sinh(b)} \left[ \frac{1}{a - b} \sinh((a - b)\tau + b) \right]_0^t$$

$$= \frac{1}{\sinh(a) - \sinh(b)} \left( \sinh((a - b)t + b) - \sinh(b) \right).$$

Let $F_{\text{cosh}}(t) = u$, where $u \sim U(0, 1)$, we have

$$u = \frac{1}{\sinh(a) - \sinh(b)} \left( \sinh((a - b)t + b) - \sinh(b) \right)$$

$$u \left( \sinh(a) - \sinh(b) \right) = \sinh((a - b)t + b) - \sinh(b)$$

$$\sinh((a - b)t + b) = \sinh(b) + u \left( \sinh(a) - \sinh(b) \right)$$

$$(a - b)t + b = \text{arcsinh} \left( \sinh(b) + u \left( \sinh(a) - \sinh(b) \right) \right).$$

Hence

$$t = \frac{1}{a - b} \left[ \text{arcsinh} \left( \sinh(b) + u \left( \sinh(a) - \sinh(b) \right) \right) - b \right]. \tag{42}$$

Therefore, we can first sample $u \sim \mathcal{U}(0, 1)$ and then get $t$ using Eq. 42.

### E.2 SAMPLING FROM UNIFORM-POWER DISTRIBUTION

**The sampling strategy.** We use the hierarchical sampling (or conditional sampling) method to sample from uniform-power distribution. We first sample a variable $u$ from uniform distribution and then sample $s$ based on $u$ and another uniform variable $v$:

    i) Sample $u \sim U(0, 1)$

    ii) Sample $v \sim U(0, 1)$

    iii) Compute $s = u \cdot \left( 1 - (1 - v)^{1/\nu} \right)$

The name 'uniform-power distribution' is descriptive of how the samples are generated.

Then we derive that the sampling strategy leads to the PDF

$$p_{\text{up}}(s; \nu) = \nu \int_s^1 \frac{1}{\tau} \left( 1 - \frac{s}{\tau} \right)^{\nu - 1} d\tau$$

The formula for $s$ can be seen as a product of two terms: $u$ and a term depending on $v$. We first define a new intermediate variable $x$ by

$$x = 1 - (1 - v)^{1/\nu}$$

and find the probability distribution of $x$. Since $x$ is a function of a random variable $v \sim U(0, 1)$, we can find its distribution using the change of variables technique, specifically by first finding its

CDF $F_X(x)$.

$$F_X(x) = P(X \le x) = P\left(1 - (1-v)^{1/\nu} \le x\right)$$

$$= P\left(1 - x \le (1-v)^{1/\nu}\right)$$

$$= P\left((1-x)^\nu \le 1 - v\right)$$

$$= P\left(v \le 1 - (1-x)^\nu\right)$$

Since $v \sim U(0,1)$, its CDF is $F_V(v) = v$ for $v \in [0,1]$. Therefore, the probability is simply the value of the upper bound:

$$F_X(x) = 1 - (1-x)^\nu, \quad \text{for } x \in [0,1]$$

Its PDF $p_X(x)$ is

$$p_X(x) = \frac{d}{dx} F_X(x) = \frac{d}{dx}\left(1 - (1-x)^\nu\right) = -\nu(1-x)^{\nu-1}(-1) = \nu(1-x)^{\nu-1},$$

which is the PDF of a *Beta distribution*, specifically $X \sim \text{Beta}(1, \nu)$.

The joint PDF of $u$ and $x$ is

$$p(u, x) = p_U(u) \cdot p_X(x) = 1 \cdot \nu(1-x)^{\nu-1} = \nu(1-x)^{\nu-1}$$

for the domain $u \in [0,1]$ and $x \in [0,1]$.

In order to find the marginal PDF $p(s)$, we perform a change of variables from $(u, x)$ to $(s, u)$, where $s = ux$. The inverse transformation is $(u, s) \to (u, x = s/u)$.

The joint PDF $p(s, u)$ in the new coordinate system is given by:

$$p(s, u) = p(u, x(s, u)) \left|\det(J)\right|$$

where $J$ is the Jacobian of the inverse transformation.

$$J = \begin{pmatrix} \frac{\partial u}{\partial s} & \frac{\partial u}{\partial u} \\ \frac{\partial x}{\partial s} & \frac{\partial x}{\partial u} \end{pmatrix} = \begin{pmatrix} 0 & 1 \\ \frac{1}{u} & -\frac{s}{u^2} \end{pmatrix}$$

The determinant of the Jacobian is $\det(J) = 0 \cdot (-\frac{s}{u^2}) - 1 \cdot \frac{1}{u} = -\frac{1}{u}$. The absolute value is $|\det(J)| = \frac{1}{u}$.

Substituting this into the formula for $p(s, u)$:

$$p(s, u) = p(u, s/u) \cdot \frac{1}{u} = \nu\left(1 - \frac{s}{u}\right)^{\nu-1} \cdot \frac{1}{u}$$

The domain of validity for this joint distribution is defined by $0 \le u \le 1$ and $0 \le x \le 1$. Substituting $x = s/u$, the second condition becomes $0 \le s/u \le 1$, which implies $0 \le s \le u$. Thus, the domain for $(s, u)$ is the triangular region $0 \le s \le u \le 1$.

Finally, to get the marginal PDF $p(s)$, we integrate the joint PDF $p(s, u)$ over all possible values of $u$. For a fixed value of $s$, $u$ must range from $s$ to 1.

$$p(s) = \int_s^1 p(s, u)\, du = \int_s^1 \frac{\nu}{u}\left(1 - \frac{s}{u}\right)^{\nu-1} du$$

By renaming the integration variable from $u$ to $\tau$, we get:

$$p(s) = \nu \int_s^1 \frac{1}{\tau}\left(1 - \frac{s}{\tau}\right)^{\nu-1} d\tau$$

**Preserving the CU task when $s = t_0$.** Our strategy of first sampling $s$ and then $t$ is a straightforward approach that ensures the CU task is preserved when $s \to t_0$. For example, consider the sampling strategy where $s$ is sampled first, followed by $t$ drawn uniformly from the interval $[s, 1]$. The conditional probability density function (PDF) of $t$ given $s$ is

$$p(t|s) = \frac{1}{1-s}, \quad t \in [s, 1].$$

In the limit as $s$ approaches zero, this conditional distribution gracefully converges to the standard uniform distribution on $[0, 1]$:

$$\lim_{s\to 0} p(t|s) = \lim_{s\to 0} \frac{1}{1-s} = 1, \quad \text{which is the PDF of } \mathcal{U}(0,1).$$

This contrasts with the alternative strategy used by Zhou et al. (2025), where the sampling order is reversed: first $t \sim \mathcal{U}(0,1)$, and then $s \sim \mathcal{U}(0,t)$. To find the conditional distribution $p(t|s)$ in this case, we must first compute the marginal distribution $p(s)$.

$$p(s) = \int_s^1 p(s|t)p(t)\, dt$$

$$= \int_s^1 \frac{1}{t} \cdot 1\, dt$$

$$= -\ln(s).$$

Applying Bayes' rule then gives the conditional PDF:

$$p(t|s) = \frac{p(s|t)p(t)}{p(s)}$$

$$= \frac{1/t}{-\ln(s)} = \frac{1}{-t\ln(s)}, \quad t \in [s, 1].$$

As $s \to 0$, all the probability mass of this distribution becomes infinitely concentrated at $t = 0$. The limiting distribution is therefore not uniform, but rather the Dirac delta distribution centered at the origin:

$$\lim_{s\to 0} p(t|s) = \delta(t).$$

## F    DETAILS ON TIME-AWARE MMD LOSS

The MMD loss used in IMM (Zhou et al., 2025) is

$$\mathcal{L}_{\text{IMM}}(\theta) = \frac{1}{B/M} \sum_{i=1}^{B/M} w(s^i, t^j) \frac{1}{M^2} \sum_{j=1}^{M} \sum_{k=1}^{M} k(\boldsymbol{g}_\theta(\boldsymbol{x}_{t^i}^{(i,j)}, t^i, s^i), \boldsymbol{g}_\theta(\boldsymbol{x}_{t^i}^{(i,k)}, t^i, s^i))$$

$$+ k(\boldsymbol{g}_{\theta^-}(\boldsymbol{x}_{r^i}^{(i,j)}, r^i, s^i), \boldsymbol{g}_{\theta^-}(\boldsymbol{x}_{r^i}^{(i,k)}, r^i, s^i)) - 2k(\boldsymbol{g}_\theta(\boldsymbol{x}_{t^i}^{(i,j)}, t^i, s^i), \boldsymbol{g}_{\theta^-}(\boldsymbol{x}_{r^i}^{(i,k)}, r^i, s^i)). \tag{43}$$

Our proposed time-aware kernel is redefined as

$$k((\boldsymbol{x}_1, t_1, s_1), (\boldsymbol{x}_2, t_2, s_2)) = -\|\boldsymbol{x}_1 - \boldsymbol{x}_2\|_2 \cdot e^{-\frac{\|t_1 - t_2\|_2 + \|s_1 - s_2\|_2}{\sigma}}, \tag{44}$$

where the parameter $\sigma$ controls the sensitivity to time differences. We use the negative Euclidean distance kernel rather than the Gaussian-like kernel applied in Zhou et al. (2025), since changing the kernel function will change the effective loss weighting function analyzed in Section 3.3. With the time-aware kernel Eq. 44, the loss becomes

$$\mathcal{L}_{\text{tMMD}}(\theta) = \frac{1}{B/M} \sum_{i=1}^{B/M} \frac{1}{M^2} \sum_{j=1}^{M} w(t^{(i,j)}, s^{(i,j)}) \sum_{k=1}^{M} -k($$

$$(\boldsymbol{g}_\theta(\boldsymbol{x}_{t^{(i,j)}}, t^{(i,j)}, s^{(i,j)}), t^{(i,j)}, s^{(i,j)}), (\boldsymbol{g}_\theta(\boldsymbol{x}_{t^{(i,k)}}, t^{(i,k)}, s^{(i,k)}), t^{(i,k)}, s^{(i,k)}))$$

$$- k((\boldsymbol{g}_{\theta^-}(\boldsymbol{x}_{r^{(i,j)}}, r^{(i,j)}, s^{(i,j)}), r^{(i,j)}, s^{(i,j)}), (\boldsymbol{g}_{\theta^-}(\boldsymbol{x}_{r^{(i,k)}}, r^{(i,k)}, s^{(i,k)}), r^{(i,k)}, s^{(i,k)}))$$

$$+ 2k((\boldsymbol{g}_\theta(\boldsymbol{x}_{t^{(i,j)}}, t^{(i,j)}, s^{(i,j)}), t^{(i,j)}, s^{(i,j)}), (\boldsymbol{g}_{\theta^-}(\boldsymbol{x}_{r^{(i,k)}}, r^{(i,k)}, s^{(i,k)}), r^{(i,k)}, s^{(i,k)})), \tag{45}$$

that is,

$$\mathcal{L}_{\text{tMMD}}(\theta) = \frac{1}{B/M} \sum_{i=1}^{B/M} \frac{1}{M^2} \sum_{j=1}^{M} w(t^{(i,j)}, s^{(i,j)}) \sum_{k=1}^{M} -\|\boldsymbol{g}_\theta(\boldsymbol{x}_{t^{(i,j)}}, t^{(i,j)}, s^{(i,j)})$$

$$- \boldsymbol{g}_\theta(\boldsymbol{x}_{t^{(i,k)}}, t^{(i,k)}, s^{(i,k)})\|_2 \cdot e^{-\frac{\|t^{(i,j)} - r^{(i,k)}\|_2 + \|s^{(i,j)} - s^{(i,k)}\|_2}{\sigma}}$$

$$- \|\boldsymbol{g}_{\theta^-}(\boldsymbol{x}_{r^{(i,j)}}, r^{(i,j)}, s^{(i,j)}) - \boldsymbol{g}_{\theta^-}(\boldsymbol{x}_{r^{(i,k)}}, r^{(i,k)}, s^{(i,k)})\|_2 \cdot e^{-\frac{\|t^{(i,j)} - r^{(i,k)}\|_2 + \|s^{(i,j)} - s^{(i,k)}\|_2}{\sigma}}$$

$$+ 2\|\boldsymbol{g}_\theta(\boldsymbol{x}_{t^{(i,j)}}, t^{(i,j)}, s^{(i,j)}) - \boldsymbol{g}_{\theta^-}(\boldsymbol{x}_{r^{(i,k)}}, r^{(i,k)}, s^{(i,k)})\|_2 \cdot e^{-\frac{\|t^{(i,j)} - r^{(i,k)}\|_2 + \|s^{(i,j)} - s^{(i,k)}\|_2}{\sigma}}. \tag{46}$$

Our proposed objective remains a valid distribution-level loss. Instead of considering distributions over the state space $\mathbb{R}^D$, we now consider distributions over the joint space of states and times, $\mathbb{R}^D \times \mathbb{R} \times \mathbb{R}$. Let a sample from this joint space be denoted by $\boldsymbol{z} = (\boldsymbol{x}, t, s)$. Our kernel in Eq. equation 44 is a valid kernel function $k(\boldsymbol{z}_1, \boldsymbol{z}_2)$ on this augmented space. Our time-aware kernel reformulates the IMM loss in a way that preserves its core objective and theoretical guarantees while offering a practical advantage. The exponential term acts as a soft weighting mechanism: pairs of samples with vastly different time coordinates contribute less to the loss, while pairs with similar time coordinates are compared more directly.

## G IMPLEMENTATION DETAILS

### G.1 EXPERIMENTAL SETTINGS ON CIFAR-10

**Model architecture and training details.** Unless otherwise specified, we use the DDPM++ architecture (Karras et al., 2022). The dropout rate is set to $0.2$. All models are trained using the RAdam (Liu et al., 2019) optimizer with a learning rate of $0.0001$ for 400K iterations. For the optimizer, $\beta_1$ and $\beta_2$ are set as 0.9 and 0.999, and $\epsilon$ is set as $10^{-8}$. Except for Table 4, where a batch size of 1024 is used, the batch size is set to 512 for all other experiments unless otherwise noted. The EMA decay rate is set to 0.9999. For faster training, we use TF32 precision.

**Loss function implementation.** For the particle-level $\ell_2$ norm loss, we implement the objective defined in Eq. 39. For our proposed time-aware MMD loss, we use the formulation in Eq. 40. Finally, when comparing against the MMD loss from Zhou et al. (2025), we use their original formulation as specified in Eq. 5. While mathematically equivalent, their performance may differ in practice due to implementation-level factors such as numerical precision.

**Sampling configuration.** Following Zhou et al. (2025), we define the effective time range from $t_0 = 0.006$ (image) to $t_T = 0.994$ (Gaussian noise). For 2-step generation, the restart sampler is used for CU models, while the pushforward sampler is used for VU models. A detailed description of these samplers can be found in Zhou et al. (2025). In both cases, the intermediate time point is set to $t_1 = 0.6$.

**Discretization curriculum details.** When applying the discretization curriculum (Song & Dhariwal, 2023), we initialize the parameter $\epsilon$ in $h(t)$ to 0.5 and halve this value every 50K training iterations.

### G.2 EXPERIMENTAL SETTINGS ON IMAGENET

**Fine-tuning on ImageNet-64$\times$64.** To ensure a fair comparison, we conduct experiments using the official ECT codebase[2]. We adopt their setting, which includes pre-training on a diffusion model and applying a discretization curriculum to accelerate convergence since the training period is very short (only $0.39\%$ of iCT). First, we reproduce the results from the ECT paper using their prescribed 100K training iterations. Our reproduced results are slightly poorer than those reported, which we attribute to hardware differences (their used H100 GPUs are different from ours). For our method, we use the CU (constant-upper-limit) variant for stable transfer. We make two minimal modifications to the loss function, motivated by the short fine-tuning duration on a pre-trained model:

- Gamma Weighting Function ($t^\gamma e^{-\lambda t}$): We adjusted the parameters from $\gamma = 1, \lambda = 6$ (used for training CIFAR-10 from scratch) to $\gamma = 1, \lambda = 3$.

- Time Sampling PDF: We switched from a sine distribution to a uniform distribution.

These changes ensure a more even distribution of weights across all time intervals, which is crucial for effective learning in a very short training window.

**Training-from-scratch on latent-space ImageNet-256$\times$256.** We use the unofficial PyTorch implementation[3] of MeanFlow, which faithfully reproduces the paper's results (the official code is not

---

[2]https://github.com/locuslab/ect/tree/imgnet
[3]https://github.com/zhuyu-cs/MeanFlow

in PyTorch). Here, we use our VU (variable-upper-limit) variant. To align with the MeanFlow baseline, we use BF16 precision and adjust the $\epsilon$ value from $0.005$ (for CIFAR-10) to $0.02$. MeanFlow's use of model guidance makes training the $s = t$ case crucial since model guidance is calculated using $\boldsymbol{f}_{\theta^-}(\boldsymbol{x}_t, t, t)$. Therefore, for simplicity and a fair comparison, we partially adopt their sampling strategy: we set the probability of the $s = t$ case to $0.75$ using their log-normal distribution $(\mathrm{lognorm}(-0.4, 1.0))$, while retaining our own sampling method for all other cases where $s \neq t$. For our loss function, we intuitively set the auxiliary intensity $\nu = 1$ to handle the potential instability of the latent space. Our only hyperparameter tuning is on the Gamma parameter; we find that weighting the loss more heavily near $t = t_0$ is beneficial for latent-space ImageNet, so we set $\gamma = 0.2$. We do not tune the classifier-free guidance (CFG) settings, which could potentially yield further improvements.

## H  ADDITIONAL RESULTS

In this sections, we provide additional quantitative results.

### H.1  DATA AUGMENTATION

Different from EDM (Karras et al., 2022), we disable EDM augmentation (Karras et al., 2022) and enable dataset xflip, which we find improves the convergence speed when using batch size $512$, as shown in Table 8. When scaling the batch size to $1024$, we enable EDM augmentation (Karras

Table 8: Effect of EDM augmentation and dataset xflip with batch size $512$ in 400K iterations.

| EDM Augmentation | Dataset xflip | 1-step | 2-step |
|:---:|:---:|:---:|:---:|
| ✓ | ✗ | 7.42 | 5.10 |
| ✗ | ✗ | 6.53 | 4.41 |
| ✗ | ✓ | 6.40 | 4.28 |

et al., 2022) and keep dataset xflip open. The effect of EDM augmentation with batchsize $1024$ is illustrated in Table 9.

Table 9: Effect of EDM augmentation with batch size $1024$ in 400K iterations.

| EDM Augmentation | Dataset xflip | 1-step | 2-step |
|:---:|:---:|:---:|:---:|
| ✗ | ✓ | 2.74 | 2.11 |
| ✓ | ✓ | 2.63 | 1.93 |

### H.2  ABLATION STUDIES FROM CU TO VU

In Table 3, we establish the final VU variant from the naïve baseline. Here we report the results of VU under the CU baseline D, with results presented in Table 10.

Table 10: Evaluation of improvements on the variable-upper-limit (VU) variant starting from the CU baseline.

| Configuration | 1-step | 2-step |
|:---|:---:|:---:|
| D Baseline | 3.02 | 2.24 |
| M + $s$-sampling | 2.94 | 2.15 |
| K + $t$-sampling | 2.91 | 2.15 |
| L + Time-aware MMD loss | 2.86 | 2.11 |

## H.3 IMPROVEMENTS OF GCM FOR ECT

We also conduct a set of ablation experiments starting from the strong ECT (Geng et al., 2024) baseline. We first replicate the ECT framework (translating its EDM-based components to the equivalent Flow Matching interpolant used in our paper for consistency, and the details can be referred to the TrigFlow framework in Lu & Song (2024), Appendix B) without diffusion pre-training to ensure a fair comparison in the training-from-scratch scenario. To maintain consistency with our other ablations, these experiments use a batch size of 512 and are trained for 400K iterations. The results of progressively applying our improvements to the ECT baseline are summarized in Table 11. These results lead to two key conclusions: i) our final configuration (e, f) significantly outperforms the strong ECT baseline (a) in the training-from-scratch setting, confirming the effectiveness of our proposed modules, and ii) The performance drop from (a) to (b) highlights ECT's heavy reliance on a carefully tuned discretization curriculum. In contrast, our method achieves strong performance without requiring this potentially resource-intensive curriculum tuning.

Table 11: Evaluating the enhancements to ECT achieved by our CU framework.

| Configuration | 1-step | 2-step |
|---|---|---|
| a) ECT (w/o Diffusion Pretraining) | 3.87 | 2.84 |
| b) - Discretization Curriculum | 4.63 | 3.09 |
| c) - ECT Discretization + Exp Discretization | 4.38 | 3.09 |
| d) + Time Preconditioning | 4.38 | 3.05 |
| e) - ECT Loss Weighting & Time Sampling + Ours | 2.91 | 2.20 |
| f) + Time-aware MMD Loss | 2.91 | 2.19 |

# I  VISUALIZATIONS

We provide visualizations of our generated samples on CIFAR-10 generated by EDM Karras et al. (2022) model and ImageNet-64×64 dataset generated by EDM2-XL Karras et al. (2024) model. All visualized samples are randomly generated.

On CIFAR-10, for the CU model, the 1-step and 2-step results are presented in Fig. 5 and Fig. 6, respectively. Correspondingly, results for the VU model are shown in Fig. 7 (1-step) and Fig. 8 (2-step).

On ImageNet-64×64, only CU variant is trained since it is finetuned from the diffusion model EDM2 (Karras et al., 2024) with only one time input. The 1-step and 2-step samples are shown in Fig. 9 and Fig. 10, respectively.

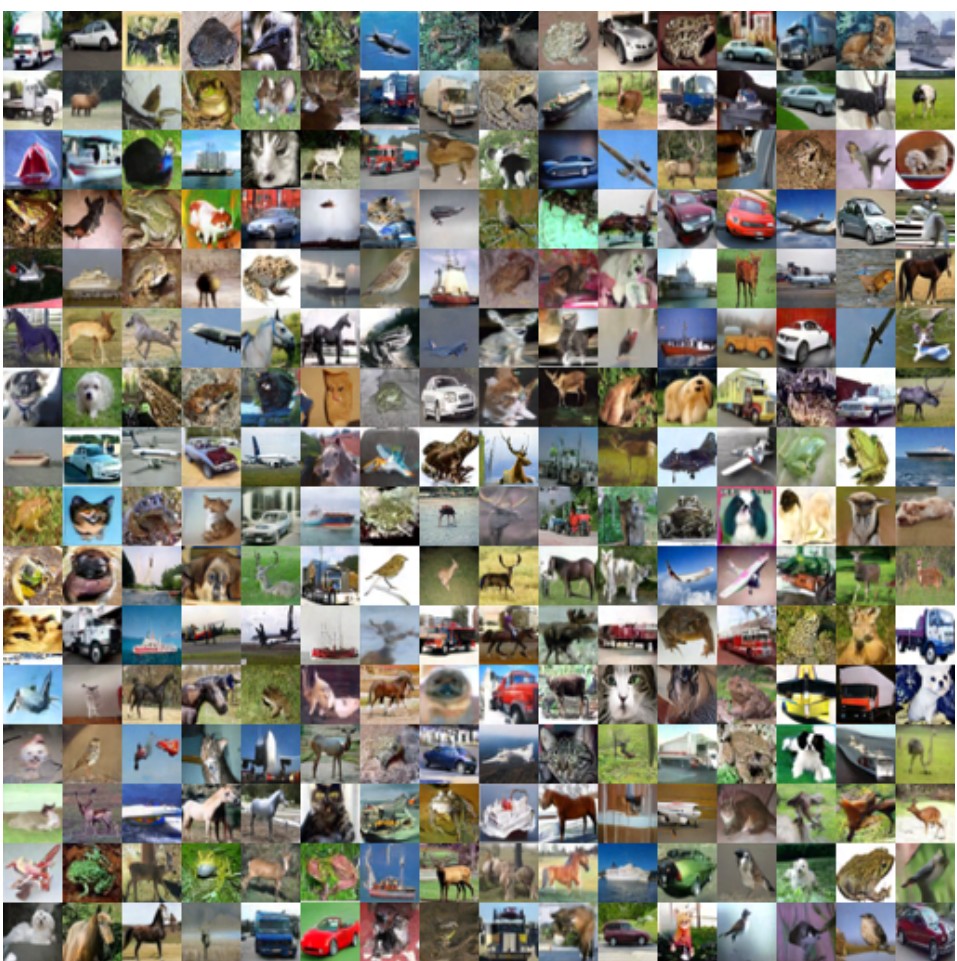

Figure 5: The 1-step results of unconditional CIFAR-10 with CU (FID=2.59).

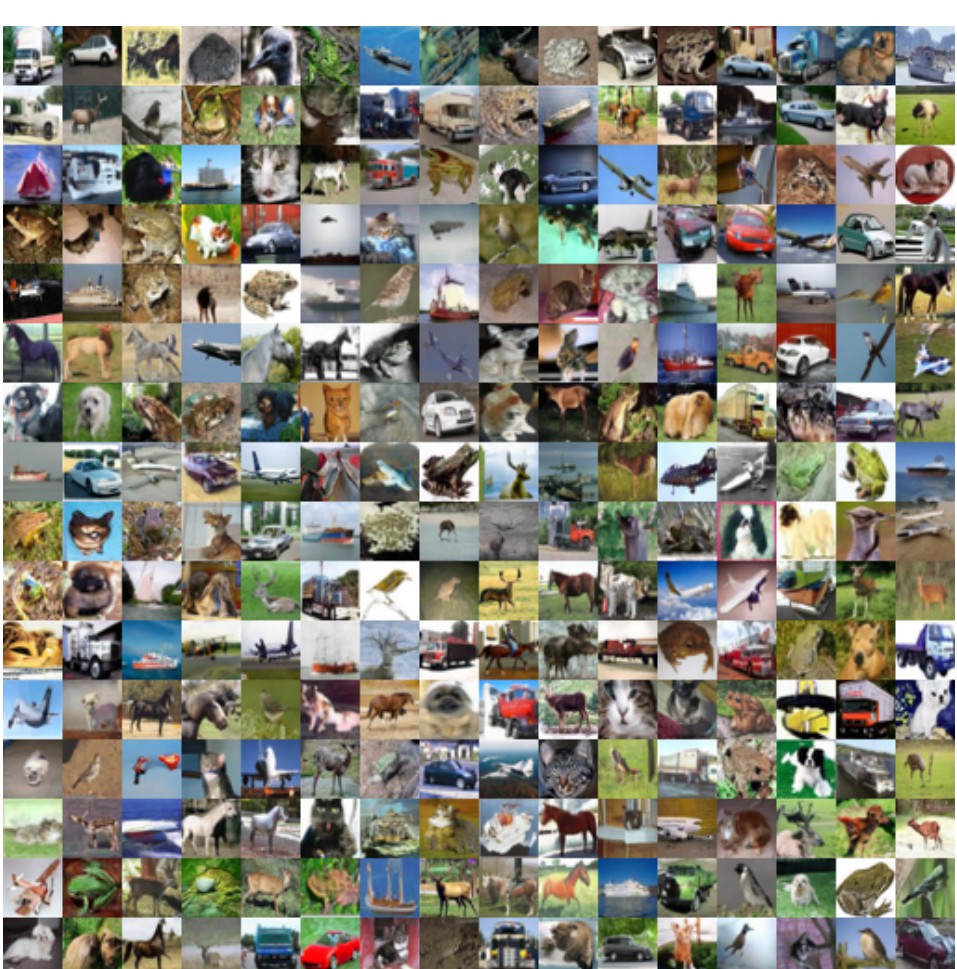

Figure 6: The 2-step results of unconditional CIFAR-10 with CU (FID=2.05).

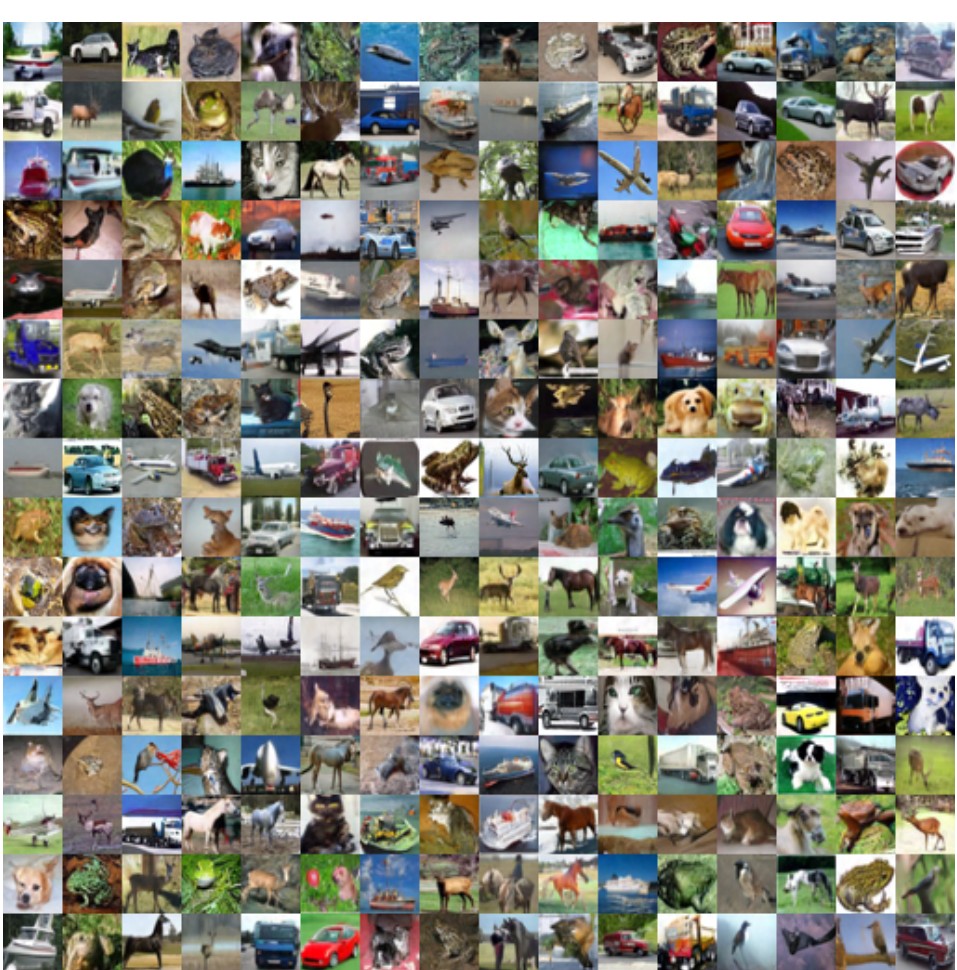

Figure 7: The 1-step results of unconditional CIFAR-10 with VU (FID=2.53).

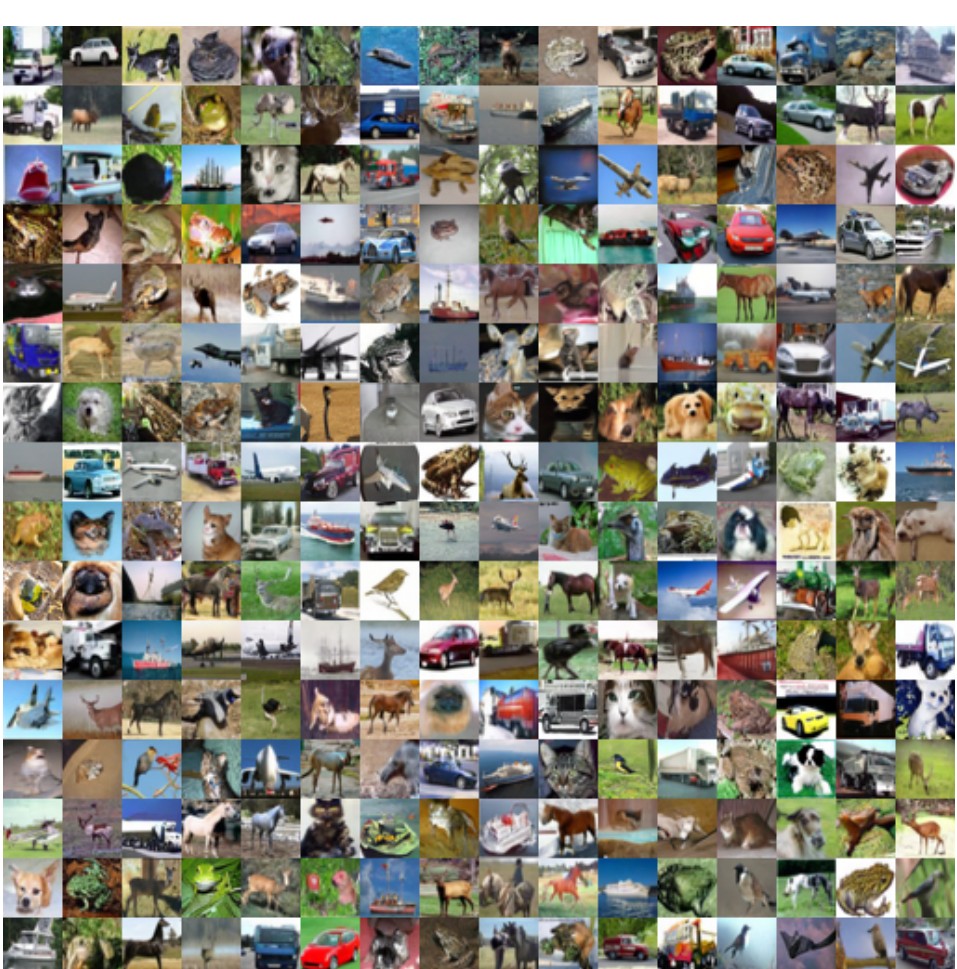

Figure 8: The 2-step results of unconditional CIFAR-10 with VU (FID=1.92).

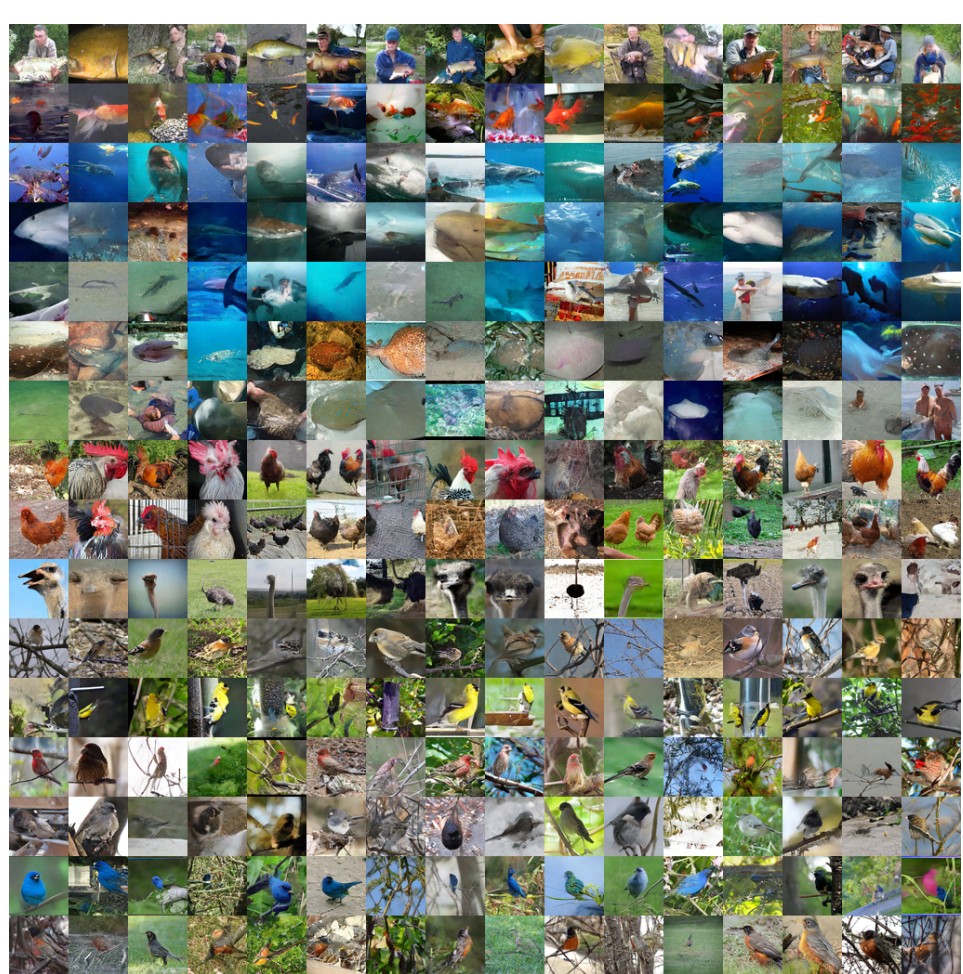

Figure 9: The 1-step results of conditional ImageNet-64×64 with CU (EDM2-XL, FID=3.44).

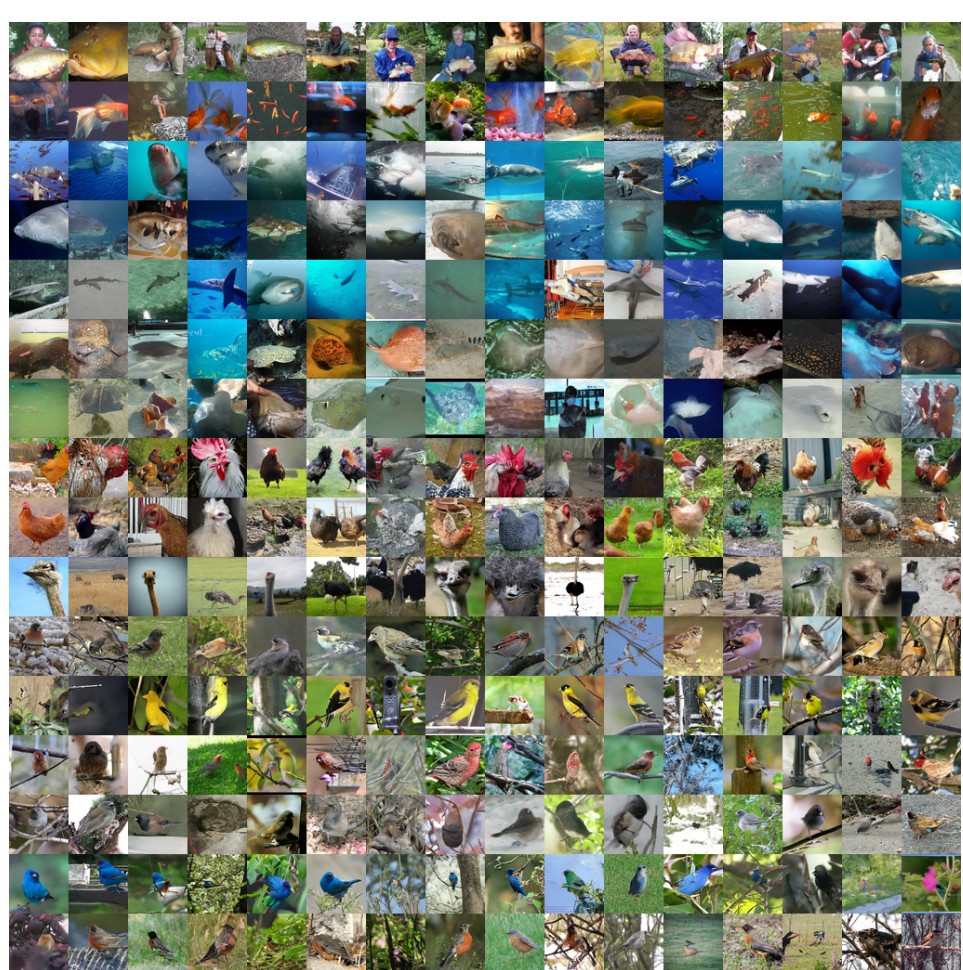

Figure 10: The 2-step results of conditional ImageNet-64×64 with CU (EDM2-XL, FID=1.98).

