# OpenReview forum: "A Guide to Training Consistency Models"
_ICLR.cc/2026/Conference — Submitted to ICLR 2026_

### Official Review · Reviewer_XHRg · 2025-10-24

**Soundness:** 2
**Presentation:** 2
**Contribution:** 2
**Rating:** 4
**Confidence:** 4

**Summary:**

This paper proposes several training techniques to improve the sample quality of consistency models, including time step discretization, time preconditioning, time weighting and distribution, etc. The authors evaluate their method on CIFAR-10.

**Strengths:**

Improving the discrete consistency model (CM)’s sample quality is an important topic, as it does not require Jacobian-vector products (JVPs). Continuous-time CMs rely on JVPs, which are not well supported by ML libraries such as PyTorch and thus add additional implementation burden.

**Weaknesses:**

My biggest concern is that the paper starts from a naive baseline, without considering already well-established training techniques. For example, the baseline (A) starts from a 1-step FID of 6.40, which is significantly higher than existing baselines such as iCT and ECT. Therefore, although Table 2 shows that their techniques improve performance from this naive baseline, it fails to show that it improves over iCT or ECT. The only comparison against these baselines is in the final result table, which does not show the effect of each design choice proposed in this paper when applied to those baselines.

Next, I list the points regarding each proposed component:

* **Time step discretization:** The motivation behind the proposed discretization is that it minimizes the overall discretization error coming from finite difference approximation. However, in Table 6(j), the continuous-time model (Analytic) underperforms the discrete CM. Doesn't this show that the discretization error does not matter when the discrete CM employs a sufficiently small step size?
* **Weighting function:** The reasoning behind w(t) is not convincing. Why do we want to match the gradient magnitude of CM and diffusion models? They are optimizing different losses with different purposes. Also, the diffusion model’s gradient magnitude depends on specific instances. For example, once it achieves a minimum, the expected gradient will be zero.
* Did you mean once you plug (12) into (11), the norm of (11) is the same as the norm of (10)? Why?
* **Auxiliary variable-upper-limit integration task:** ECT’s time-step upper bound is zero. See [https://github.com/locuslab/ect/blob/4311059770f54821d151a9b0e1f76770a5f3930e/training/loss.py#L45-L50](https://github.com/locuslab/ect/blob/4311059770f54821d151a9b0e1f76770a5f3930e/training/loss.py#L45-L50). It’s not clear if we need this technique given that the baseline already does not use the time upper bound.
* **MMD loss:** The performance improvement is marginal (in Table 2, it does not improve at all). I’d suggest removing it and instead using that space to explain some contents in Appendix C.1.

Finally, as noted by the authors, one important limitation is that the paper only evaluates on CIFAR-10. Although I understand that the ImageNet-64 experiments may be beyond the authors’ available resources, it is also the case that empirical findings in CIFAR-10 often do not transfer to ImageNet or larger datasets. One way to address this is to start from a pre-trained diffusion model and show ablations with relatively small batch sizes, as done in ECT.

**Questions:**

* Can you set A in Table 2 to ECT/iCT and show the effects of your techniques from there?
* In Table 6(b), why is linear decrease performing similarly to the proposed exponential decrease?
* ECT uses a sigmoid multiplier on CIFAR-10. See [https://github.com/locuslab/ect/blob/4311059770f54821d151a9b0e1f76770a5f3930e/training/loss.py#L45-L50](https://github.com/locuslab/ect/blob/4311059770f54821d151a9b0e1f76770a5f3930e/training/loss.py#L45-L50). Can you compare against this in Table 6(b)?

---

> ### Author Response · Authors · 2025-11-20
> **Response to Reviewer XHRg (1/4)**
>
> We sincerely thank the reviewer for their detailed, constructive feedback and for raising several critical points that have helped us significantly improve our manuscript. We address each of your concerns and questions below.
>
> ---
>
> > **Weakness 1 (biggest concern):** The paper starts from a naive baseline, without considering already well-established training techniques. For example, the baseline (A) starts from a 1-step FID of 6.40, which is significantly higher than existing baselines such as iCT and ECT.
>
> This is a critical point, and we thank you for raising it. Our motivations for the two-pronged approach (starting from a na\"ive baseline and comparing against SOTA) are as follows:
>
> i) **Clarity through Modularity:** The primary motivation for starting with a naive baseline was to provide a transparent, modular analysis. This ground-up approach allows us to clearly isolate and demonstrate the precise contribution of each proposed component (e.g., discretization, loss weighting). This can be obscured when starting from a highly-optimized, entangled baseline where the interplay of components is complex.
>
> ii) **New Experiments from a Strong Baseline (ECT):** We agree that demonstrating improvement over an established method is crucial. To address this, we have conducted a **new set of ablation experiments starting from a strong ECT baseline**. Since there is no official implementation for iCT, we chose ECT. We first replicated the ECT framework (translating its EDM-based components to the equivalent Flow Matching interpolant used in our paper for consistency, and the details can be referred to the TrigFlow framework in the sCM paper, Appendix B) without diffusion pre-training to ensure a fair comparison in the training-from-scratch scenario.
>
> To maintain consistency with our other ablations, these experiments use a batch size of 512 and are trained for 400K iterations. The results of progressively applying our improvements to the ECT baseline are summarized below:
>
> | Configuration                                  | 1-step | 2-step |
> |:-----------------------------------------------|:------:|:------:|
> | a) ECT (w/o Diffusion Pretraining)             |  3.87  |  2.84  |
> | b) - Discretization Curriculum                 |  4.63  |  3.09  |
> | c) - ECT Discretization + Exp Discretization   |  4.38  |  3.09  |
> | d) + Time Preconditioning                      |  4.38  |  3.05  |
> | e) - ECT Loss Weighting & Time Sampling + Ours |  2.91  |  2.20  |
> | f) + Time-aware MMD Loss                       |  2.91  |  2.19  |
>
> These results lead to two key conclusions:
> 1.  Our final configuration (e, f) significantly outperforms the strong ECT baseline (a) in the training-from-scratch setting, confirming the effectiveness of our proposed modules.
> 2.  The performance drop from (a) to (b) highlights ECT's heavy reliance on a carefully tuned discretization curriculum. In contrast, our method achieves strong performance **without** requiring this potentially resource-intensive curriculum tuning.
>
> We have added the new experimental results in Appendix H.3 in the revised paper.

---

> ### Author Response · Authors · 2025-11-20
> **Response to Reviewer XHRg (2/4)**
>
> ---
>
> > **Weakness 2:** Questions regarding each proposed component.
>
> **i) Time step discretization:** The motivation [...] is that it minimizes the overall discretization error [...]. However, in Table 6(j), the continuous-time model (Analytic) underperforms the discrete CM. Doesn't this show that the discretization error does not matter when the discrete CM employs a sufficiently small step size?
>
> We attribute this performance difference to the distinct practical behaviors of the analytic (continuous-time) and finite-difference (discrete-time) formulations.
> *   **Practical Instability of Analytic Methods:** As noted by prior work (e.g., Figure 1 (b) in iCT paper, analysis in sCM paper), the analytic version, which relies on `torch.jvp` for Jacobian-vector products, is often more sensitive to implementation details (like time embeddings) and requires additional stabilization techniques. This inherent practical instability can overshadow the theoretical benefits of avoiding discretization error.
> *   **Limits of Small Step Sizes:** While theoretically discretization error vanishes as the step size approaches zero, practical implementations are constrained by **numerical precision**. Excessively small step sizes can lead to numerical instability and performance degradation, as shown in related works (e.g., Figure 1 in ECT, Figure 5(c) in sCM). Our ablation studies were conducted at a sweet spot for the step size $\epsilon$, balancing approximation accuracy and numerical stability.
>
> **ii) Weighting function:** Why do we want to match the gradient magnitude of CM and diffusion models? [...] Did you mean once you plug (12) into (11), the norm of (11) is the same as the norm of (10)?
>
> Thank you for requesting this clarification.
> *   **The Shared Integration Process:** During inference, a diffusion model numerically integrates the velocity field $E[z-x|x_t]$ to generate a sample. The core objective of consistency model training is to learn the result of this exact same integration process. Given that both DM inference and CM training are fundamentally concerned with integrating the same velocity field, we hypothesize that an effective loss weighting designed to modulate errors across the integration path for a DM is directly applicable to training a CM. We design this weighting, $w(t)$, as a function of time $t$ only, making it a statistical control (like ELBO weighting) rather than an instance-level one.
>
> *   **Magnitude Control:** To be precise, our goal is not to equate the full gradient norms, but to align their dominant scalar components. We observe that while the parameter gradient ($\nabla_\theta f\_\theta$) tends to be stable (otherwise the training is easy to collapse), the error term ($E\_\\text{CM}$) can be highly volatile. Prior methods (e.g., Pseudo-Huber loss in iCT, adaptive weighting in sCM) stabilize it by normalizing this error term, making the gradient term $\\frac{E\_\\text{CM}}{\\|E\_\\text{CM}\\|_2}\nabla\_\theta f\_\theta$ (and so do $\frac{E\_\\text{DM}}{\\|E\_\\text{DM}\\|_2}\nabla\_\theta v\_\theta$) have a roughly uniform magnitude across time. This means that the overall gradient magnitude is now dominated by the external scalar weighting function. We can therefore transfer the benefits of a high-performance DM strategy by aligning these dominant scalar functions. We achieve this by setting our CM weight $w\_\text{CM}(t) \propto \\|E\_\text{DM}\\|_2 w\_\text{DM}(t)$, ensuring our weighting's temporal distribution matches that of a ELBO-weighted DM.
>
> **iii) Auxiliary variable-upper-limit integration task:** ECT’s time-step upper bound is zero. It’s not clear if we need this technique...
>
> We investigate the auxiliary VU task for two primary reasons:
> 1.  **Training Stability:** As discussed in Section 3.4, the VU task introduces a data-only, model-independent target ($u_t$) into the loss function when $s=t$. This helps stabilize the training process, which is beneficial for complex datasets.
> 2.  **Enabling Deterministic Sampling:** The VU framework enables the use of **pushforward sampling**, a deterministic method that strictly follows the ODE trajectory. This contrasts with the stochastic restart sampling used in CU models. As shown empirically in both the IMM paper and our own results, pushforward sampling consistently yields better performance for multi-step (>1) generation (see IMM Appendix, Algs. 4 & 5 for detailed defination of pushforward and restart samping).
>
> **iv) MMD loss:** The performance improvement is marginal... I’d suggest removing it and instead using that space to explain some contents in Appendix C.1.
>
> Thank you for this suggestion. We agree that while the time-aware MMD loss is a conceptual improvement, its empirical gains are modest. In the revised manuscript, we have followed your recommendation to shorten this section and used the freed space to move the key theoretical intuitions from Appendix C.1 into the main paper for better clarity.

---

> > ### Author Response · Authors · 2025-11-20
> > **Response to Reviewer XHRg (3/4)**
> >
> > ---
> >
> > > **Weakness 3:** The transferability to larger datasets like ImageNet.
> >
> > Thank you for this valuable suggestion. To demonstrate the transferability of our method, we have conducted new experiments on **ImageNet-64** and **ImageNet-256** (in the latent space).
> >
> > **i) Experiments on ImageNet-64 (Fine-tuning from a Pre-trained Model)**
> >
> > To ensure a fair comparison, we conducted experiments using the official ECT codebase. We adopted their setting, which includes pre-training on a diffusion model and applying a discretization curriculum to accelerate convergence since the training period is very short (only $0.39\\%$ of iCT).
> >
> > First, we reproduced the results from the ECT paper using their prescribed $100$K training iterations. Our reproduced results are slightly poorer than those reported, which we attribute to hardware differences (we used 8x A800 GPUs, whereas the authors used 4/8x H100 GPUs).
> >
> > For our method, we used the CU (constant-upper-limit) variant for stable transfer. We made two minimal modifications to the loss function, motivated by the short fine-tuning duration on a pre-trained model:
> > 1.  **Gamma Weighting Function ($t^\gamma e^{-\lambda t}$):** We adjusted the parameters from $\gamma=1, \lambda=6$ (used for training CIFAR-10 from scratch) to $\gamma=1, \lambda=3$.
> > 2.  **Time Sampling PDF:** We switched from a sine distribution to a uniform distribution.
> >
> > These changes ensure a more even distribution of weights across all time intervals, which is crucial for effective learning in a very short training window.
> >
> > | Model | 1-step FID | 2-step FID |
> > | :--- | :---: | :---: |
> > | ECT-S (paper) | 5.51 | 3.18 |
> > | ECT-S (reproduced) | 5.83 | 3.47 |
> > | **Ours-S** | **5.25** | **3.45** |
> > | ECT-XL (paper) | 3.35 | 1.96 |
> > | ECT-XL (reproduced) | 3.56 | 2.10 |
> > | **Ours-XL** | **3.44** | **1.98** |
> >
> > **ii) Experiments on Latent-Space ImageNet-256 (Training from Scratch)**
> >
> > We also tested our method on the more challenging task of training from scratch in the latent space of ImageNet-256. We used the unofficial PyTorch implementation of MeanFlow, which faithfully reproduces the paper's results (the official code is not in PyTorch).
> >
> > Here, we used our VU (variable-upper-limit) variant. To align with the MeanFlow baseline, we used BF16 precision and adjusted the $\epsilon$ value from $0.005$ (for CIFAR-10) to $0.02$. MeanFlow's use of model guidance makes training the $s=t$ case crucial since model guidance is calculated using $f(x_t,t,t)$. Therefore, for simplicity and a fair comparison, we partially adopted their sampling strategy: we set the probability of the $s=t$ case to $0.75$ using their log-normal distribution (lognorm($-0.4$, $1.0$)), while retaining our own sampling method for all other cases where $s\neq t$. For our loss function, we intuitively set the auxiliary intensity $\nu=1$ to handle the potential instability of the latent space. Our only significant hyperparameter tuning was on the Gamma parameter; we found that weighting the loss more heavily near $t=t_0$ was beneficial for latent-space ImageNet, so we set $\gamma=0.2$. We did not tune the classifier-free guidance (CFG) settings, which could potentially yield further improvements.
> >
> > | Model | Epochs | 1-step FID |
> > | :--- | :---: | :---: |
> > | MeanFlow-B/4 (paper) | 80 | 15.53 (Table 1f) |
> > | MeanFlow-B/4 (reproduced) | 80 | 15.43 |
> > | **Ours-B/4** | **80** | **14.27** |
> > | MeanFlow-XL/2 (paper) | 40 | ~7.8 (Figure 4) |
> > | MeanFlow-XL/2 (reproduced) | 40 | 7.75 |
> > | **Ours-XL/2** | **40** | **7.12** |
> >
> > **In summary,** these new experiments demonstrate that our method:
> > 1.  Successfully adapts to a fine-tuning scenario on **ImageNet-64**, outperforming ECT, a method specifically optimized for this setup, with minimal hyperparameter tuning.
> > 2.  Effectively transfers to training from scratch on the challenging **latent-space ImageNet-256**, achieving superior results compared to a highly tuned baseline with only minor adjustments.
> >
> > We have added the new experimental results in Section 4.1 in the revised paper.

---

> > > ### Author Response · Authors · 2025-11-20
> > > **Response to Reviewer XHRg (4/4)**
> > >
> > > ---
> > >
> > > > **Question 1:** Can you set A in Table 2 to ECT/iCT and show the effects of your techniques from there?
> > >
> > > Please see the response to Weakness 1.
> > >
> > > ---
> > >
> > > > **Question 2:** In Table 6(b), why is linear decrease performing similarly to the proposed exponential decrease?
> > >
> > > Our theoretical analysis in Section 3.2, which derives the exponential decreasing function, relies on certain simplifying assumptions to make the problem of minimizing global error tractable. In practice, while these assumptions guide us to the generally correct **shape** (i.e., a decreasing function), the exact optimal form may differ slightly.
> > >
> > > The strong performance of other decreasing functions, like the linear one, suggests that the general principle, i.e., allocating smaller step sizes (higher precision) to the low-noise region near clean data, is more critical than adhering to a precise exponential form. As shown in Table 6(b), decreasing functions as a group consistently outperform other functional shapes, which validates our core analysis.
> > >
> > > ---
> > >
> > > > **Question 3:** ECT uses a sigmoid multiplier on CIFAR-10. Can you compare against this in Table 6(b)?
> > >
> > > Thank you for the suggestion. We have added ECT's discretization schedule (which uses a sigmoid multiplier) to the comparison in Table 6(b). The updated results are below:
> > >
> > > | Discretization h(t) | 1-step | 2-step |
> > > |:--------------------|:------:|:------:|
> > > | Constant            |  4.62  |  2.50  |
> > > | EDM's               |  3.29  |**2.24**|
> > > | ECT's               |  3.20  |  2.27  |
> > > | IMM's               |  3.35  |  2.43  |
> > > | Linear Decrease     | 3.08 | 2.26 |
> > > | Quadratic Decrease  |  3.34  |  2.43  |
> > > | Cubic Decrease      |  3.46  |  2.65  |
> > > | Quadratic Increase  |  9.09  |  3.08  |
> > > | Quadratic Concave   |  3.38  |  2.28  |
> > > | Quadratic Convex    |  6.83  |  2.75  |
> > > | Exponential Decrease|**3.03**| 2.26 |
> > >
> > > And we have also added the ECT's function curve in Table 6(a), and the derivation of ECT's discretization function in Appendix C.2 in the revised manuscript.

---

### Official Review · Reviewer_Y58a · 2025-10-27

**Soundness:** 2
**Presentation:** 2
**Contribution:** 2
**Rating:** 4
**Confidence:** 4

**Summary:**

The paper proposes several techniques for improving the traning of consistency models and flow-map consistency models. By progressively adding modular improvements over a simple baseline, the paper clearly identifies the benefits and advantages of each added component. The result is a clear guideline on how to imprve consistency training, and the model achieve competitive results on CIFAR-10.

**Strengths:**

The resulting method is relatively simple, especially compared to other consistency model training procedures. Most proposed components are well motivated and justified, and verified with extensive empirical evaluation. The final results on CIFAR-10 are competitive.

**Weaknesses:**

As also mentioned by the authors, the method is only evaluated on CIFAR-10. This makes it hard to understand if the proposed improvements translate to other datasets. In addition, it would be very interesting to see how directly the various design choices work on different datasets and settings, i.e. how much additional tuning is required to get this method to work on other datasets. Ideally, it would be a great selling point if the design choices translate well with minimal tuning, but from the manuscript it is hard to tell. While I understand that not every lab has access to much computational resources, the authors could have used latent imagenet 256 x 256 which has a latent resolution of 32x32x4, with the smallest DiT settings, which I believe shouldn't surpass by much the GPU requirements of CIFAR-10. Another weakness is that the authors did not provide the code, which makes it hard to verify some of the claims made in the paper (see questions).

**Questions:**

- The baseline seems too good to be true to me. From my understanding, that's a finite difference approximation of a continuous consistency model with a constant small $dt$, trained from scratch, without weighting functions and with uniform time distribution, with the l2 loss and without discretization curricula. That alone gets 6.4 1-step FID, which is already much better than the original CT. Could the authors please provide insights on this?
- The loss in equation 5 reduces to simple l2 loss when s=0. I wonder how did the authors manage to get such good results with a simple L2 loss, while most papers need to use pseudo-huber loss or variants.
- Do you use float32 precision?

---

> ### Author Response · Authors · 2025-11-20
> **Response to Reviewer Y58a (1/2)**
>
> We thank the reviewer for their positive feedback and for raising several important questions. We have addressed each point below.
>
> ---
>
> > **Weakness 1:** As also mentioned by the authors, the method is only evaluated on CIFAR-10. This makes it hard to understand if the proposed improvements translate to other datasets. [...] It would be a great selling point if the design choices translate well with minimal tuning.
>
> Thank you for this valuable suggestion. To demonstrate the transferability of our method, we have conducted new experiments on **ImageNet-64** and **ImageNet-256** (in the latent space). Our findings show that our design choices translate effectively with minimal tuning.
>
> **i) Experiments on ImageNet-64 (Fine-tuning from a Pre-trained Model)**
>
> To ensure a fair comparison, we conducted experiments using the official ECT codebase. We adopted their setting, which includes pre-training on a diffusion model and applying a discretization curriculum to accelerate convergence since the training period is very short (only $0.39\\%$ of iCT).
>
> First, we reproduced the results from the ECT paper using their prescribed $100$K training iterations. Our reproduced results are slightly poorer than those reported, which we attribute to hardware differences (we used 8x A800 GPUs, whereas the authors used 4/8x H100 GPUs).
>
> For our method, we used the CU (constant-upper-limit) variant for stable transfer. We made two minimal modifications to the loss function, motivated by the short fine-tuning duration on a pre-trained model:
> 1.  **Gamma Weighting Function ($t^\gamma e^{-\lambda t}$):** We adjusted the parameters from $\gamma=1, \lambda=6$ (used for training CIFAR-10 from scratch) to $\gamma=1, \lambda=3$.
> 2.  **Time Sampling PDF:** We switched from a sine distribution to a uniform distribution.
>
> These changes ensure a more even distribution of weights across all time intervals, which is crucial for effective learning in a very short training window.
>
> | Model | 1-step FID | 2-step FID |
> | :--- | :---: | :---: |
> | ECT-S (paper) | 5.51 | 3.18 |
> | ECT-S (reproduced) | 5.83 | 3.47 |
> | **Ours-S** | **5.25** | **3.45** |
> | ECT-XL (paper) | 3.35 | 1.96 |
> | ECT-XL (reproduced) | 3.56 | 2.10 |
> | **Ours-XL** | **3.44** | **1.98** |
>
> **ii) Experiments on Latent-Space ImageNet-256 (Training from Scratch)**
>
> We also tested our method on the more challenging task of training from scratch in the latent space of ImageNet-256. We used the unofficial PyTorch implementation of MeanFlow, which faithfully reproduces the paper's results (the official code is not in PyTorch).
>
> Here, we used our VU (variable-upper-limit) variant. To align with the MeanFlow baseline, we used BF16 precision and adjusted the $\epsilon$ value from $0.005$ (for CIFAR-10) to $0.02$. MeanFlow's use of model guidance makes training the $s=t$ case crucial since model guidance is calculated using $f(x_t,t,t)$. Therefore, for simplicity and a fair comparison, we partially adopted their sampling strategy: we set the probability of the $s=t$ case to $0.75$ using their log-normal distribution (lognorm($-0.4$, $1.0$)), while retaining our own sampling method for all other cases where $s\neq t$. For our loss function, we intuitively set the auxiliary intensity $\nu=1$ to handle the potential instability of the latent space. Our only significant hyperparameter tuning was on the Gamma parameter $\gamma$; we found that weighting the loss more heavily near $t=t_0$ was beneficial for latent-space ImageNet, so we set $\gamma=0.2$. We did not tune the classifier-free guidance (CFG) settings, which could potentially yield further improvements.
>
> | Model | Epochs | 1-step FID |
> | :--- | :---: | :---: |
> | MeanFlow-B/4 (paper) | 80 | 15.53 (Table 1f) |
> | MeanFlow-B/4 (reproduced) | 80 | 15.43 |
> | **Ours-B/4** | **80** | **14.27** |
> | MeanFlow-XL/2 (paper) | 40 | ~7.8 (Figure 4) |
> | MeanFlow-XL/2 (reproduced) | 40 | 7.75 |
> | **Ours-XL/2** | **40** | **7.12** |
>
> **In summary,** these new experiments demonstrate that our method:
> 1.  Successfully adapts to a fine-tuning scenario on **ImageNet-64**, outperforming ECT, a method specifically optimized for this setup, with minimal hyperparameter tuning.
> 2.  Effectively transfers to training from scratch on the challenging **latent-space ImageNet-256**, achieving superior results compared to a highly tuned baseline with only minor adjustments.
>
> We have added the new experimental results in Section 4.1 in the revised paper.

---

> > ### Author Response · Authors · 2025-11-20
> > **Response to Reviewer Y58a (2/2)**
> >
> > ---
> >
> > > **Weakness 2:** Another weakness is that the authors did not provide the code, which makes it hard to verify some of the claims made in the paper.
> >
> > We are committed to reproducibility. To ensure that our work can be fully verified and built upon, we will release our source code and pre-trained model weights upon the paper's publication.
> >
> > ---
> >
> > > **Question 1 & 2:** The baseline seems too good to be true to me. [...] That alone gets 6.4 1-step FID, which is already much better than the original CT. The loss in equation 5 reduces to simple l2 loss when s=0. I wonder how did the authors manage to get such good results with a simple L2 loss, while most papers need to use pseudo-huber loss or variants.
> >
> > The reviewer is correct that our loss function uses a finite difference approximation with a constant small $dt$, trained from scratch, without weighting functions, with uniform time distribution, and without discretization curricula. However, the loss function is the **$\ell_2$ norm ($\\|\\cdot\\|_2$)**, not the **$\ell_2$ loss** (i.e., the squared $\ell_2$ norm, $\\|\\cdot\\|_2^2$).
> >
> > The $\ell_2$ norm is, in fact, a special case of the Pseudo-Huber loss (as detailed in line 126-128). We found this distinction to be critical for performance. When we trained the same baseline using a standard squared $\ell_2$ loss, the performance was very poor (FID > $20$). The use of the $\ell_2$ norm (or, more generally, a Pseudo-Huber loss) was vital for stabilizing training and achieving the strong baseline result, a finding that is consistent with observations made in the iCT paper. This choice is a primary reason why our baseline is significantly better than the original CT.
> >
> > ---
> >
> > > **Question 3:** Do you use float32 precision?
> >
> > Our experiments were conducted using **TF32 precision**. We built our work on the publicly available IMM codebase, which defaults to this setting. This provided a significant training speedup (approximately $50\\%$ on CIFAR-10) compared to standard FP32 precision.

---

> ### Comment · Reviewer_Y58a · 2025-11-25
>
> I thank the authors for answering my questions. After reading through the replies and the answers to the other reviewers, I am inclined to raise my score, given also the strong performance shown on the additional experiments.
>
> However, I remain a bit skeptical of the very good results obtained with such a simple method, especially for the baseline. It's still hard for me to believe that the used loss, pseudo-huber with c=0, is enough for achieving good results with the simple baseline. The request for sharing the code during the reviewing and rebuttal period was to be able to verify what the authors did in practice. Would the authors be able to share even just a simple colab notebook whit the baseline implementation, and potentially the possibility to run a simple test on cifar10?

---

> ### Author Response · Authors · 2025-11-26
> **Response to Reviewer Y58a**
>
> **Dear Reviewer,**
>
> We sincerely thank you for your positive feedback and your inclination to raise the score.
>
> We completely understand your skepticism regarding the strong performance achieved by such a simple baseline (Pseudo-Huber loss with $c=0$). To fully address this and facilitate verification, we have included the source code containing a minimal implementation of our naive baseline (mainly in `training/loss.py`) based on the IMM codebase in the **updated supplementary material (ZIP file)**.
>
> To reproduce the results, please follow these steps:
>
> 1.  **Environment Setup:** Please follow the installation instructions in the official IMM codebase (https://github.com/lumalabs/imm).
> 2.  **Dataset Preparation:** Please prepare the CIFAR-10 dataset following the instructions provided in the IMM codebase, and **place the processed dataset in the `./datasets/` directory**.
> 3.  **Training:** Please configure your WandB `api_key` and `entity` in `configs/cifar10.yaml`. Then, you can launch the training using the following command (note that we used 8 GPUs in our experiments):
>     ```bash
>     WANDB_SILENT="true" bash run_train.sh 8 cifar10.yaml
>     ```
>
> **Expected Results for Quick Verification:**
> To help verify the progress, we would like to note that the model typically achieves a 1-step FID of approximately **11 at 100k iterations** and **8 at 200k iterations**.
>
> We hope this helps verify our findings in practice. Please let us know if you have any further questions regarding the code.
>
> Best regards,
>
> The Authors

---

> ### Comment · Reviewer_Y58a · 2025-11-28
>
> I thank the authors for providing the code. I could verify the implementation and run some tests. I decided to raise my score to 6.

---

> > ### Author Response · Authors · 2025-11-28
> >
> > Dear Reviewer,
> >
> > Thank you very much for your message. We are delighted to hear that you were able to verify our implementation with the provided code. We sincerely appreciate you taking the time to run the tests and for your decision to raise the score.
> >
> > Best regards,
> >
> > The Authors

---

### Official Review · Reviewer_RfR2 · 2025-11-03

**Soundness:** 2
**Presentation:** 3
**Contribution:** 2
**Rating:** 6
**Confidence:** 4

**Summary:**

Recent methods for training consistency models rely on numerous design choices, often set empirically, that significantly impact performance. This paper empirically investigates the design choices for the  critical components of training consistency models and suggests best practices for training consistency models. These components include discretization schedule, time preconditioning, time sampling, loss weighting,  targets in loss function, as well as the expression for the loss function itself which in this case, is inductive moment matching style distributional loss. The analysis is done on CIFAR-10, and the resulting model gets competitive FID on CIFAR-10 for 1-step and 2-step sampling.

**Strengths:**

1. Consistency models have been widely adapted to distill diffusion models or train 1-step or 2-step from scratch. The detailed empirical analysis in this paper is useful for the research community as it provides guidelines in training such models in practice.
2. The main paper has been written in an easy to understand manner and various design decisions have been explained clearly (though the proofs in the appendix can be simplified).

**Weaknesses:**

1. Results and analysis is done only on CIFAR-10 which is quite small. It would have been useful to show that the configuration from CIFAR-10 can be transferred to IMageNet-64.
2. Errors in Time Preconditioning and typos in some other expressions:
- The general form for time preconditioning  $c(t) = K \int \dfrac{dt}{h(t)} + C$. Upon substituting $h(t) = \epsilon e^{ - \mu t}$, we get $c(t) = \dfrac{K}{\epsilon \mu} e^{\mu t} + C$. This function is exponentially decreasing when it should be exponentially increasing to get $c’(t)h(t)$ as constant. This expression is also different from the one in the paper $c(t) = \dfrac{e^{-\mu t} - 1}{\mu}$.
-  In line 1142 and line 1147: After taking the limit, $r$ should be replaced with $t$ in the expression.
3. From Table 2, the role of introducing some of the components is unclear. For instance, time-aware MMD loss has similar 1-step performance and only marginal gain (+0.01 FID) on 2 step over the other design choices. The recommended time preconditioning seems to result in worse 1-step performance. In Table 3 for variable-upper-limit, the granular analysis done for constant-upper limit is missing. Therefore, it is difficult to understand the contributions of different components like discretization function, time preconditioning, loss weighting etc. for variable upper limit case.

**Questions:**

1. What time preconditioning was used in practice? The recommended expression from time conditioning differs from the one that can be derived as per the reasoning in the paper.

---

> ### Author Response · Authors · 2025-11-20
> **Response to Reviewer RfR2 (1/2)**
>
> We thank the reviewer for their constructive feedback and insightful comments. We have addressed each point below and have incorporated corresponding revisions and new experimental results into our manuscript.
>
> ---
>
> > **Weakness 1:** Results and analysis is done only on CIFAR-10 which is quite small. It would have been useful to show that the configuration from CIFAR-10 can be transferred to ImageNet-64.
>
> Thank you for this valuable suggestion. To demonstrate the transferability of our method, we have conducted new experiments on **ImageNet-64** and **ImageNet-256** (in the latent space).
>
> **i) Experiments on ImageNet-64 (Fine-tuning from a Pre-trained Model)**
>
> To ensure a fair comparison, we conducted experiments using the official ECT codebase. We adopted their setting, which includes pre-training on a diffusion model and applying a discretization curriculum to accelerate convergence since the training period is very short (only $0.39\\%$ of iCT).
>
> First, we reproduced the results from the ECT paper using their prescribed $100$K training iterations. Our reproduced results are slightly poorer than those reported, which we attribute to hardware differences (we used 8x A800 GPUs, whereas the authors used 4/8x H100 GPUs).
>
> For our method, we used the CU (constant-upper-limit) variant for stable transfer. We made two minimal modifications to the loss function, motivated by the short fine-tuning duration on a pre-trained model:
> 1.  **Gamma Weighting Function ($t^\gamma e^{-\lambda t}$):** We adjusted the parameters from $\gamma=1, \lambda=6$ (used for training CIFAR-10 from scratch) to $\gamma=1, \lambda=3$.
> 2.  **Time Sampling PDF:** We switched from a sine distribution to a uniform distribution.
>
> These changes ensure a more even distribution of weights across all time intervals, which is crucial for effective learning in a very short training window.
>
> | Model | 1-step FID | 2-step FID |
> | :--- | :---: | :---: |
> | ECT-S (paper) | 5.51 | 3.18 |
> | ECT-S (reproduced) | 5.83 | 3.47 |
> | **Ours-S** | **5.25** | **3.45** |
> | ECT-XL (paper) | 3.35 | 1.96 |
> | ECT-XL (reproduced) | 3.56 | 2.10 |
> | **Ours-XL** | **3.44** | **1.98** |
>
> **ii) Experiments on Latent-Space ImageNet-256 (Training from Scratch)**
>
> We also tested our method on the more challenging task of training from scratch in the latent space of ImageNet-256. We used the unofficial PyTorch implementation of MeanFlow, which faithfully reproduces the paper's results (the official code is not in PyTorch).
>
> Here, we used our VU (variable-upper-limit) variant. To align with the MeanFlow baseline, we used BF16 precision and adjusted the $\epsilon$ value from $0.005$ (for CIFAR-10) to $0.02$. MeanFlow's use of model guidance makes training the $s=t$ case crucial since model guidance is calculated using $f(x_t,t,t)$. Therefore, for simplicity and a fair comparison, we partially adopted their sampling strategy: we set the probability of the $s=t$ case to $0.75$ using their log-normal distribution (lognorm($-0.4$, $1.0$)), while retaining our own sampling method for all other cases where $s\neq t$. For our loss function, we intuitively set the auxiliary intensity $\nu=1$ to handle the potential instability of the latent space. Our only significant hyperparameter tuning was on the Gamma parameter; we found that weighting the loss more heavily near $t=t_0$ was beneficial for latent-space ImageNet, so we set $\gamma=0.2$. We did not tune the classifier-free guidance (CFG) settings, which could potentially yield further improvements.
>
> | Model | Epochs | 1-step FID |
> | :--- | :---: | :---: |
> | MeanFlow-B/4 (paper) | 80 | 15.53 (Table 1f) |
> | MeanFlow-B/4 (reproduced) | 80 | 15.43 |
> | **Ours-B/4** | **80** | **14.27** |
> | MeanFlow-XL/2 (paper) | 40 | ~7.8 (Figure 4) |
> | MeanFlow-XL/2 (reproduced) | 40 | 7.75 |
> | **Ours-XL/2** | **40** | **7.12** |
>
> **In summary,** these new experiments demonstrate that our method:
> 1.  Successfully adapts to a fine-tuning scenario on **ImageNet-64**, outperforming ECT, a method specifically optimized for this setup, with minimal hyperparameter tuning.
> 2.  Effectively transfers to training from scratch on the challenging **latent-space ImageNet-256**, achieving superior results compared to a highly tuned baseline with only minor adjustments.
>
> We have added the new experimental results in Section 4.1 in the revised paper.

---

> > ### Author Response · Authors · 2025-11-20
> > **Response to Reviewer RfR2 (2/2)**
> >
> > ---
> >
> > > **Weakness 2:** Errors in Time Preconditioning and typos in some other expressions.
> > > *   The general form for time preconditioning...
> > > *   In line 1142 and line 1147: After taking the limit, r should be replaced with t in the expression.
> >
> > Thank you for pointing out these errors. We sincerely apologize for the oversight.
> >
> > **i) The Form of the Time Preconditioning Function**
> >
> > You are correct. The expression for $c(t)$ in the main text (line 209) was a typo. The correct form, which was used in our loss function, is $c(t)=\frac{e^{\mu t} - 1}{\mu}$, as correctly stated in Table 1.
> >
> > The derivation for this form is based on the constraint that the function behaves like the identity near the origin, i.e., $\lim_{t \to 0} c(t)/t = 1$. This ensures that our preconditioning behaves similar to the naive choice $c(t)=t$. The derivation is as follows:
> > $$
> >     \lim_{t\to 0} \frac{c(t)}{t} = \lim_{t\to 0}\frac{K\int \frac{dt}{h(t)}+C}{t} =\lim_{t\to 0}\frac{\frac{K}{\epsilon \mu}e^{\mu t}+C}{t} =\lim_{t\to 0}\frac{\frac{K}{\epsilon \mu}(1+\mu t + O(t^2))+C}{t} =\lim_{t\to 0}\frac{K}{\epsilon} + \frac{\frac{K}{\epsilon \mu} + C}{t} + O(t) =1.
> > $$
> > For this limit to hold, the constant term must be 1 and the $\frac{1}{t}$ term must be 0. This gives us two conditions:
> > 1.  $\frac{K}{\epsilon}=1 \implies K=\epsilon$
> > 2.  $\frac{K}{\epsilon \mu} + C = 0 \implies C = -\frac{K}{\epsilon \mu} = -\frac{1}{\mu}$
> >
> > Substituting these back into the general form for $c(t)$ gives the correct expression: $c(t)=\frac{e^{\mu t} -1}{\mu}$.
> >
> > **ii) Typo in Limit Expression (r vs. t)**
> >
> > Thank you for catching this as well. You are correct that `r` should be replaced with `t` after taking the limit. We will correct both of these errors in the revised manuscript.
> >
> > ---
> >
> > > **Weakness 3:** From Table 2, the role of introducing some of the components is unclear... The recommended time preconditioning seems to result in worse 1-step performance. In Table 3 for variable-upper-limit, the granular analysis done for constant-upper limit is missing.
> >
> > Thank you for this feedback. We acknowledge that the motivation and contribution of each component could have been explained more clearly.
> >
> > **i) Time-aware MMD Loss**
> >
> > While the empirical performance gain is modest, the primary motivation for the time-aware MMD loss is **conceptual**. It extends the MMD loss from IMM by removing the time-sharing constraint within each group, which provides greater modeling flexibility.
> >
> > **ii) Role of Time Preconditioning**
> >
> > The motivation for introducing time preconditioning is to strike a better **balance between 1-step and 2-step performance**. As shown in Table 2, we observed that:
> > *   Adding only the **exponential discretization function** improved 1-step FID significantly (6.40 → 5.73) but degraded 2-step FID (4.28 → 4.40).
> > *   By incorporating our proposed **time preconditioning** on top of this, we achieved a better trade-off. The final performance (5.75 for 1-step, 4.31 for 2-step) maintains most of the 1-step gains while mitigating the harm to 2-step performance.
> >
> > **iii) Granular Analysis for the Variable-Upper-Limit (VU) Case**
> >
> > To address the reviewer's concern, we have added a new ablation study for the VU case, mirroring the analysis done for the CU case. This new table clearly demonstrates the stepwise improvements gained from each component, starting from a naive baseline. This table will replace the previous Table 3 in the main paper, and the old Table 3 will be moved to Appendix H.2.
> >
> > | Configuration | 1-step FID | 2-step FID |
> > | :--- | :---: | :---: |
> > | G Baseline + $s$-sampling | 6.59 | 4.27 |
> > | H + Discretization Function | 5.50 | 4.10 |
> > | I + Time Preconditioning | 5.49 | 4.06 |
> > | J + Loss Weighting | 2.94 | 2.15 |
> > | K + $t$-sampling | 2.91 | 2.15 |
> > | L + Time-aware MMD Loss | **2.86** | **2.11** |
> >
> > ---
> >
> > > **Question 1:** What time preconditioning was used in practice? The recommended expression from time conditioning differs from the one that can be derived as per the reasoning in the paper.
> >
> > We apologize for the confusion caused by the typo. As addressed in our response to Weakness 2, this was a typographical error in line 209 of the main text. The correct formulation used in all our experiments is the one presented in Table 1: $c(t) = \frac{e^{\mu t} - 1}{\mu}$. We will ensure this is corrected throughout the revised manuscript.

---

> > > ### Comment · Reviewer_RfR2 · 2025-11-27
> > >
> > > Thanks for addressing my concerns about time preconditioning. The new table with ablation analysis of VU case is also helpful. I also thank the authors for running additional experiments on ImageNet-64 and ImageNet-256 in the limited period of time. The overall paper is stronger now. For now I would like to retain my score which recommends acceptance.

---

> > > > ### Author Response · Authors · 2025-11-28
> > > >
> > > > Dear Reviewer,
> > > >
> > > > Thank you for your positive and encouraging feedback. We are very glad that our response addressed your concerns and that you found the new ablation analysis and the additional ImageNet experiments helpful. We sincerely appreciate your time, guidance, and your support in recommending our work for acceptance.
> > > >
> > > > Best regards,
> > > >
> > > > The Authors

---

### Meta-Review · Area_Chair_KJ2Y · 2026-01-06

**Summary:**

This paper proposes a “training playbook” for (discrete) consistency models by starting from a deliberately simple baseline and incrementally adding modules (discretization schedule, time preconditioning, loss weighting / time sampling, and a time-aware MMD-style distribution loss), supported by ablations and improved CIFAR-10 1–2 step FID.

Advantages:
* Clear, modular empirical study that isolates effects of key training components.
* Rebuttal meaningfully strengthened the submission with additional experiments beyond CIFAR-10 and clarification/correction of time-preconditioning.
* Practical guidance may help researchers reproduce strong CM results without deeply entangled pipelines.

Disadvantages:
* The work starts from a naive baseline and initially under-positions prior, stronger frameworks; the “playbook” is not cleanly presented as improvements on top of iCT/sCM/ECT as the primary starting point.
* Several modules appear low-impact or conceptually motivated but empirically marginal (e.g., time-aware MMD), and the benefit attribution remains sensitive to training protocol details.
* Reproducibility and credibility concerns (baseline strength, implementation choices) were only fully alleviated late via code release for verification, and the broader generality of the prescriptions still feels under-justified.

Consider the score changes (Y58a: 4→6) and RfR2 retaining a 6, the paper currently sits in a borderline region, making it challenging to make a clear decision. After careful examination of the discussion, despite stronger results after rebuttal (including additional ImageNet-64 and latent ImageNet-256 experiments), I recommend reject because the paper’s core framing starts from a naive baseline and treats recent, stronger CM frameworks (e.g., iCT, sCM, ECT) as optional add-ons, making it difficult to clearly assess novelty and incremental merit relative to established pipelines. More broadly, I also believe continuous-time-based formulations are likely to become the mainstream direction as forward-mode differentiation becomes increasingly well supported in practice (e.g., already in JAX and improving in PyTorch). I hope this feedback helps the authors strengthen the manuscript and consider a future resubmission.

**Reviewer Concerns:**

The rebuttal made major revisions by adding ImageNet-64 fine-tuning results (ECT comparisons) and latent ImageNet-256-from-scratch comparisons; it also corrected the time preconditioning typo and added the missing VU granular ablation, improving interpretability. However, outstanding concerns remain around the paper’s positioning and comparative clarity: the central narrative still builds from a naive baseline and only secondarily integrates stronger modern CM frameworks (iCT, sCM, ECT) as starting points, making it hard to cleanly disentangle “new contributions” from “re-discovering” best practices. Although the rebuttal added an experiment with ECT, I would highly encourage the authors to redo the analysis starting from more established frameworks.

**Reviewer Scores:**

* Y58a already changed from 4 → 6 after code was provided and verified; unlikely to change further.

* RfR2 explicitly stated they will retain the current 6 after concerns were addressed; unlikely to change.

* XHRg is the most plausible to move slightly upward given the new ECT-starting ablations and ImageNet results, but their central critique (starting from a naive baseline and ignoring iCT/sCM/ECT as the natural starting point) is only partially mitigated, so a large score increase is unlikely.

Given these changes, the paper would obtain an overall borderline rating.

---

### Decision · Program_Chairs · 2026-01-26

Reject